# Dual interference with host neuropeptide signaling allows parasitoid wasp to hijack host sugar metabolism

Zhi-Zhi Wang [iD] [1,2,3,4,5], Ruo-Fei Ma[1,2,3,5], Li-Cheng Gu[1,2,3], Li-Zhi Wang[1,2,3], Ting Chen[1,2,3], Pei Yang[1,2,3], Jia-Ni Zou[1,2,3], Jiang-Yan Zhu[1,2,3], Zhi-Wei Wu[1,2,3], Yue-Nan Zhou [iD] [1,2,3], Min Shi[2,3], Xing-Xing Shen[1,2,3], Jian-Hua Huang [iD] [1,2,3] & Xue-Xin Chen [iD] [1,2,3,4 ✉]

## Abstract

**Changes in host carbohydrate metabolism determine the outcome of host–parasite relationships, but the underlying mechanistic basis remains elusive. Here, we show that the parasitoid wasp *Cotesia vestalis* induces trehalose accumulation in its host, the moth *Plutella xylostella*, largely independently of insulin/adipokinetic hormone signalling and food intake. Instead, parasitoids rewire host carbohydrate metabolism via two pathways activated by the evolutionarily conserved short neuropeptide F (sNPF), a functional analogue of mammalian neuropeptide Y. Parasitoid-derived teratocytes secrete sNPF that interacts with the sNPF receptor (sNPFR) on host cells, and contributes to host hypertrehalosemia by promoting glycogenolysis in the fat body. We further find that a parasitoid-symbiotic virus induces expression of host-encoded sNPF, which stimulates glycolysis in the host midgut. Furthermore, we show that the host sNPF-sNPFR complex stimulates $G_q/Ca^{2+}$ signalling, while the parasitoid sNPF, exhibiting higher receptor affinity, triggers $G_i/cAMP$ signalling. Molecular docking analyses suggest that the observed distinct receptor activation properties may be attributed to structural variations in the sNPF-sNPFR binding pocket. Collectively, our findings uncover an unexpected role of peripheral sNPFs in the regulation of carbohydrate metabolism during host–parasite interactions.**

**Keywords** Short Neuropeptide F; Humoral Factor; Sugar Metabolism; Host–parasite Interaction
**Subject Categories** Evolution & Ecology; Metabolism; Microbiology, Virology & Host Pathogen Interaction

## Introduction

Carbohydrates serve as the primary energy source for animals (Matsuda et al, 2015). Metabolic homeostasis enables living organisms to adapt to environmental changes. To maintain a balance between carbohydrate catabolism and synthesis, organisms coordinate systemic nutrient homeostasis through humoral factors. Insulin and counterregulatory hormones, such as glucagon, are key humoral factors regulating circulating carbohydrate homeostasis (Bansal and Wang, 2008; Ojha et al, 2019). Glucagon stimulates hepatic glycogenolysis and gluconeogenesis in response to low levels of circulating glucose, and it is used clinically for the treatment of episodic hypoglycaemia (Sloop et al, 2005). In contrast, insulin inhibits hepatic glucose production and promotes circulating carbohydrate storage (Barthel and Schmoll, 2003). Impaired insulin and glucagon activities lead to dysregulated carbohydrate metabolism. Similar to mammals, carbohydrate homeostasis in invertebrates is also substantially influenced by humoral factors. The identification and characterisation of *Drosophila melanogaster* insulin-like peptides and their glucagon equivalent, adipokinetic hormone (AKH), revealed functional conservation in metabolic homeostasis and the underlying signalling mechanisms (Kim and Rulifson, 2004; Oh et al, 2019; Rulifson et al, 2002). In addition, other humoral factors linked to carbohydrate homeostasis include GABA, NPY and or its insect functional homologue short neuropeptide F (sNPF), etc. (Grayson et al, 2013; Marks and Waite, 1997; Meng et al, 2016; Oh et al, 2019) Together, the modulation of carbohydrate homeostasis is comparatively conserved between vertebrates and invertebrates and involves a complex interplay of multiple hormonal and neural signals.

Parasitism is an ecological behaviour that is common in nature and involves interactions between two or more species (Poulin and Morand, 2000). The survival of parasites primarily relies on the nutrient supply provided by their hosts. To this end, parasites sense the host's nutrition status and evolve to adapt through genetic,

[1]Institute of Insect Sciences, College of Agriculture and Biotechnology, Zhejiang University, Hangzhou, China. [2]Ministry of Agriculture and Rural Affairs Key Lab of Molecular Biology of Crop Pathogens and Insect Pests, Zhejiang University, Hangzhou, China. [3]Key Laboratory of Biology of Crop Pathogens and Insects of Zhejiang Province, Zhejiang University, Hangzhou, China. [4]State Key Lab of Rice Biology and Breeding, Zhejiang University, Hangzhou, China. [5]These authors contributed equally: Zhi-Zhi Wang, Ruo-Fei Ma.
✉E-mail: xxchen@zju.edu.cn

epigenetic and metabolic adjustments (Mancio-Silva et al, 2017; Schneider et al, 2023; Zuzarte-Luís and Mota, 2018). In addition to directly consuming host nutrients, parasites usually modulate host physiology and metabolism to optimise their growth and development (Pennacchio et al, 2014). Changes in the content and composition of carbohydrates have been widely observed in the parasitised hosts. Notably, many parasitoid wasps, which are of agricultural and ecological importance, significantly increase the blood sugar level of their hosts during the parasitic process (Dahlman and Bradleighvinson, 1977; Dahlman and Vinson, 1975; Gade, 1991; Huang et al, 2008; Thompson, 1986; Thompson et al, 1990). Further studies revealed that the parasitisation-induced increase in host blood sugar was primarily attributed to parasitoid-derived factors, including venom, polydnavirus and teratocytes (Kryukova et al, 2021; Nakamatsu et al, 2001). Polydnavirus (PDV) is a type of specialist symbiotic virus, including Bracovirus associated with braconid wasps and Ichnovirus associated with ichneumonoid wasps, and displays multiple functions, including host immune regulation and development arrest during the host-parasitoid wasp interactions (Strand and Burke, 2014). Teratocytes are cells derived from the serosa surrounding the wasp embryo, and are released into the host's haemocoel when eggs hatch (Strand, 2014). However, little is known about the molecular mechanism that controls carbohydrate homeostasis in the host following parasitisation.

*Cotesia vestalis* (Hymenoptera: Braconidae) is a solitary endoparasitoid wasp of the diamondback moth *Plutella xylostella*, one of the most widespread lepidopteran pest species of cruciferous vegetable crops (Wang et al, 2018; You et al, 2013). Here, we discovered two sources of peripheral sNPFs that regulate sugar metabolism through distinct G protein-mediated pathways in the *C. vestalis–P. xylostella* system, shedding light on host–parasite metabolic modulation.

# Results

## Parasitism affects both host glycogenolysis and glycolysis

To determine whether *C. vestalis* parasitisation impacts the sugar metabolism of the host *P. xylostella* larvae, we analysed the concentration of carbohydrates in the haemolymph of the host larvae. Ten common sugar components of the *P. xylostella* haemolymph were detected by HPLC (Fig. 1A; Appendix Table S1). Among them, trehalose, a glucose disaccharide, was the main carbohydrate, accounting for 79.94% of the total sugars, followed by fructose (8.7%), sorbitol (5.95%), glucose (4.43%) and sucrose (0.91%). The remaining five sugars are barely detectable. We next detected changes in haemolymph sugar contents in parasitised and non-parasitised *P. xylostella* larvae at different stages, including the middle 3rd instar (3M), late 3rd instar (3L), early 4th instar (4E), middle 4th instar (4M) and late 4th instar (4L) stages (Appendix Fig. S1A). The trehalose concentration in the parasitised *P. xylostella* larvae continually increased during the parasitism process and showed increases of ~20%, 27%, 70% and 38% at 3L, 4E, 4M and 4L, respectively (Fig. 1B). The concentrations of sorbitol and fructose were increased at the early stage of parasitism and decreased at the later stage (Appendix Fig. S1B,C), whereas the

levels of glucose and sucrose remained unaffected (Appendix Fig. S1D,E). Therefore, we focused on the underlying mechanism of parasitism-induced hypertrehalosemia in subsequent research.

The circulating trehalose levels result from a well-controlled balance between enzymatic synthesis and degradation (Shukla et al, 2015). We measured the mRNA expression profiles of genes involved in trehalose metabolism (Fig. 1C). Compared with non-parasitised *P. xylostella* larvae, parasitised larvae showed increased expression of trehalose synthesis-related genes, including glycogen phosphorylase (*PxGP*), hexokinase (*PxHK*) and trehalose-6-phosphate synthase (*PxTPS*), but not phosphoglucose mutase (*PxPGM*) (Fig. 1D; Appendix Fig. S1F). The transcription and activity of trehalase (*PxTREH*), which catalyses the rate-limiting step of trehalose degradation, were not affected by *C. vestalis* parasitism (Appendix Fig. S1F,G).

*GP* and *HK* act as key rate-limiting enzymes for glycogenolysis and glycolysis, respectively, catalysing the production of glucose 6-phosphate (G6P), which is the upstream metabolite of trehalose (Tellis et al, 2023). The tissue expression pattern showed that *PxGP* and *PxTPS* were highly expressed in the fat body, whereas *PxHK* was highly expressed in the midgut; both of these tissues are important tissues for the storage, digestion, and utilisation of carbohydrates (Appendix Fig. S1H). G6P accumulation was greater in the fat body of parasitised *P. xylostella* larvae, and this increased accumulation was accompanied by a significant reduction in glycogen storage (Fig. 1E–G). We next performed RNA interference (RNAi) in the parasitised *P. xylostella* larvae to knock down the expression of *PxGP* and *PxHK*. As shown in Fig. 1H, the mRNA expression levels of *PxGP* and *PxHK* were significantly reduced by 61.3% and 71.1%, respectively, in the dsRNA-treated larvae compared with those in the control group. Strikingly, the trehalose levels in *PxGP* and *PxHK* knockdown hosts were markedly decreased by 20% and 17%, respectively (Fig. 1I). Taken together, our results suggest that *C. vestalis* parasitism might promote both glycogenolysis and glycolysis in host larvae, which would lead to an increase in the circulating trehalose levels.

## Host hypertrehalosemia is crucial for wasp offspring fitness

To clarify the significance of high levels of circulating trehalose in the host haemolymph during parasitism by *C. vestalis*, we injected ds*PxGP* into parasitised host larvae to reduce the host trehalose levels, and then evaluated the biological traits of *C. vestalis* offspring (Fig. 2A). Strikingly, we detected a significant decrease in the glycogen content but not the glucose or trehalose content of *C. vestalis* offspring pupated from ds*PxGP*- treated hosts (Fig. 2B; Appendix Fig. S2A,B), suggesting that circulating trehalose from the host haemolymph is a key source of glycogen storage for the parasitoid offspring.

The level of glycogen stored in feeding larvae always gradually decreases during metamorphosis to provide energy for pupae(-Matsuda et al, 2015). Although there was no effect on the pupation rate, head width and pupal stage duration of parasitoid offspring (Fig. 2C; Appendix Fig. S2C,D), we found that reduced trehalose concentrations had negative effects on the eclosion, longevity and fecundity. First, approximately 30% of the pupae emerging from the ds*PxGP*-treated hosts failed to eclose (Fig. 2D). Male wasps from the ds*PxGP*-treated hosts exhibited an approximately 0.5 days delay

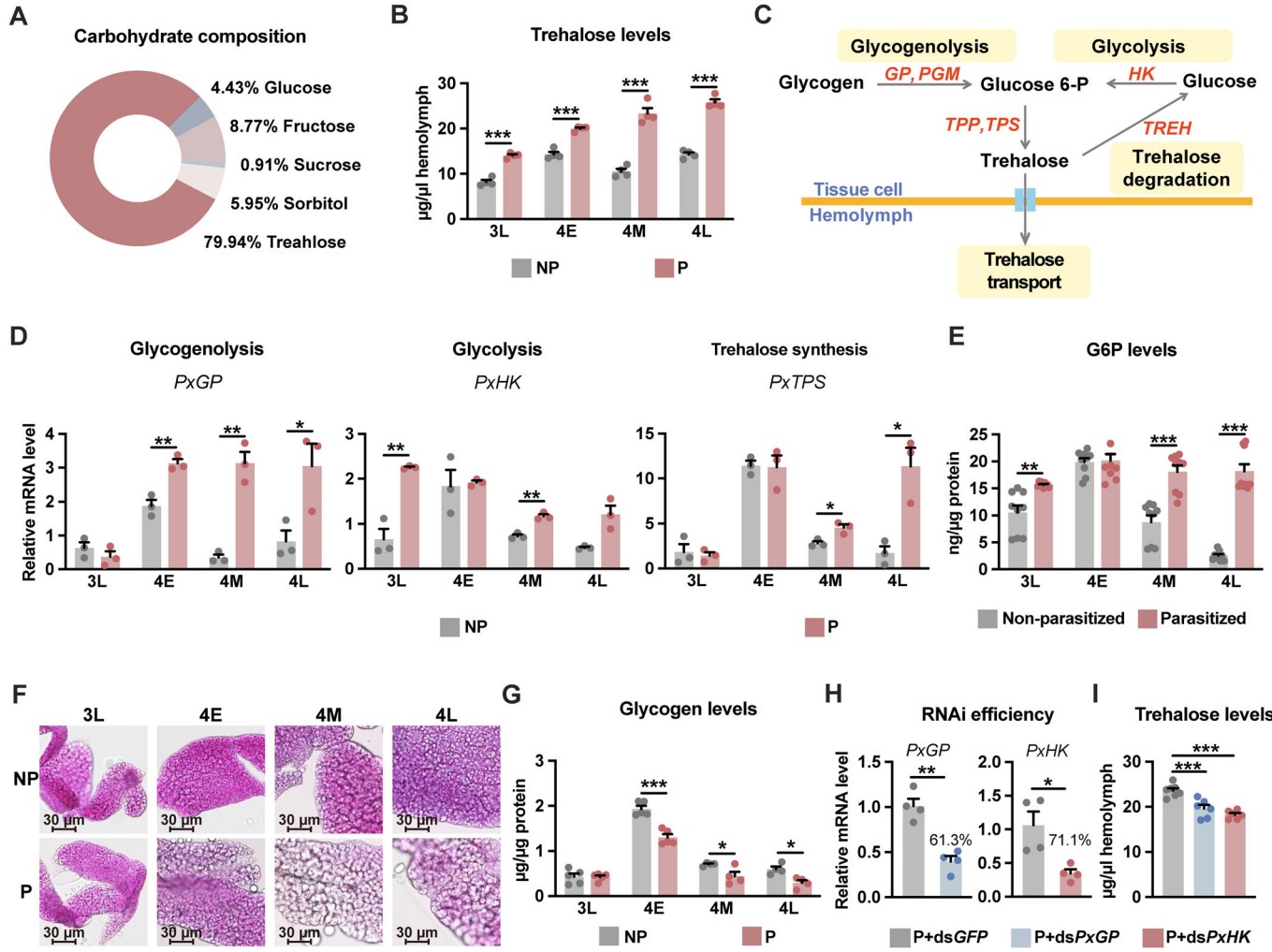

**Figure 1. Both host glycogenolysis and glycolysis are affected by parasitism.**

(A) Carbohydrate composition in the haemolymph of *P. xylostella*. (B) Circulative trehalose levels in non-parasitised (NP) or *C. vestalis*-parasitised (P) host larvae (*n* = 4). (C) Schematic diagram of the procedures for trehalose metabolism. (D) Relative mRNA levels of *PxGP*, *PxHK* and *PxTPS* (*n* = 3) from non-parasitised and *C. vestalis*-parasitised host larvae (whole body) at different instars. (E) G6P content in the fat body (*n* = 9). (F, G) PAS staining of the fat body (F) and its glycogen content (G, *n* = 4–5). Scale bars, 30 μm. (H) RNAi efficiency for the experiments in (I) was determined by relative mRNA levels of *PxGP* and *PxHK* in *C. vestalis* parasitised host larvae (whole body) at 4M instar (*n* = 4). (I) Trehalose concentration in the haemolymph of parasitised host larvae treated with ds*PxGFP*, ds*PxGP* or ds*PxHK* (*n* = 6–7). 3L, late 3rd instar; 4E, early 4th instar; 4M, middle 4th instar; 4L, late 4th instar. Data are represented as mean ± SEM. *$P < 0.05$, **$P < 0.01$, ***$P < 0.005$, not shown $P > 0.05$. The exact $P$ values are provided in Appendix Table S5. Two-tailed unpaired Student's *t* test, except for G, One-way ANOVA and Tukey's multiple comparison tests. Source data are available online for this figure.

before reaching an eclosion ratio of 50% compared with the controls (Fig. 2E). Second, the number of eggs laid per hour by a single mated female wasp decreased from 18 to 11 when host trehalose levels were reduced, whereas the fecundity of virgin wasps was not affected (Fig. 2F). To investigate the differential effects of host glycogen reduction on male and female offspring, two additional experimental groups were designed: females emerging from dsGP-treated hosts (dsGP♀) mated with males from dsGFP-treated hosts (dsGFP♂), denoted as the dsGP♀ × dsGFP♂ group, and the reciprocal cross, dsGP♂ × dsGFP♀. The dsGP♀ × dsGFP♂ group exhibited fecundity levels comparable to those of the control group. In contrast, the dsGP♂ × dsGFP♀ group showed a slight—though statistically insignificant—reduction in fecundity. Third, all of the male adult wasps emerging from the ds*PxGP*-treated hosts

died at 3 days post eclosion on a water-only diet, whereas more than 50% of the individuals in the control groups survived for one or two additional days (Fig. 2G). This negative effect on the longevity of male wasps can be compensated for by providing sugar meals (Appendix Fig. S2E). Overall, the trehalose level in host larvae had a great impact on the fitness of parasitoid offspring, especially the male offspring.

## Parasitism-induced hypertrehalosemia is largely independent of insulin-PI3K/AKT and AKH signalling

Insulin and AKH are the two crucial hormones involved in regulating carbohydrate metabolism. Insulin-phosphatidylinositol 3-kinase (PI3K) /Akt signalling regulates the storage and

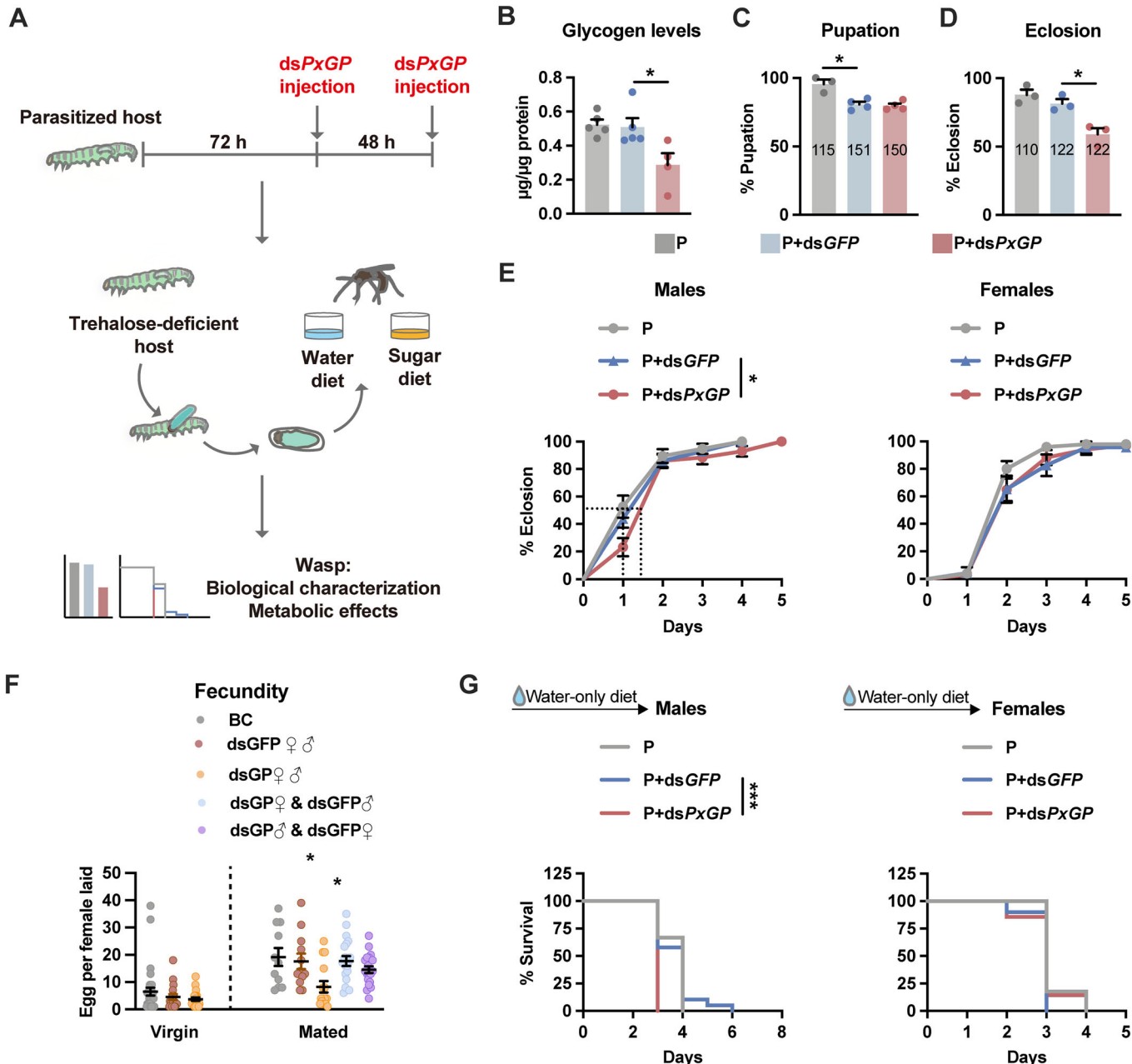

**Figure 2.  Lower host trehalose concentration impaired the fitness of *C. vestalis* progenies.**

(A) Schematic diagram of the experiment designed for (B–G) and Appendix Fig. S2. Parasitised hosts were injected with ds*PxGP* twice consecutively to generate trehalose-deficient hosts. Newly emerged wasp offspring were collected and fed with sugar (orange) or water-only (blue) and were then subjected to metabolic and biological traits measurements. (B–D) Glycogen content (B, n = 4–5), pupation rate (C, n = 3-4) and eclosion rate (D, n = 3) of wasp offspring. The total number of wasps for each treatment in (C, D) was shown on each bar graph. (E) Eclosion time of *C. vestalis* males (left, n = 38 for grey, n = 57 for blue, n = 42 for red) and females (right, n = 49 for grey, n = 22 for blue, n = 33 for red). The dot lines mark time to an eclosion ratio of 50%. (F) Number of eggs laid per hour by each virgin (left) or mated (right) female offspring. Adult parasitoid wasps (both mated and virgin) used for parasitation were fed a sugar diet for ~2 days prior to the experiment. (G) Survival curves of *C. vestalis* males (left, n = 30 for grey, n = 38 for blue, n = 44 for red) and females (right, n = 34 for grey, n = 40 for blue, n = 35 for red). Data are represented as mean ± SEM. *P< 0.05, **P < 0.01, not shown P > 0.05. The exact P values are provided in Appendix Table S5. One-way ANOVA and Tukey's multiple comparison tests, except for (E, G), Kaplan–Meier log-rank test. Source data are available online for this figure.

production of glucose, whereas AKH is involved in glycogen mobilisation (Gáliková and Klepsatel, 2023; Hopkins et al, 2020). We questioned whether parasitism impacts the insulin and AKH signalling pathways in the host (Fig. 3A). Transcriptome analysis revealed that wasp parasitisation did not affect the transcript levels

of genes involved in insulin signalling in the central nervous system (CNS), fat body, or midgut of the host, except for *insulin-like peptide 1* (*ILP1*), *ILP3* and *insulin receptor* (*InR*) (Fig. 3B). As one of the most critical factors of the insulin-PI3K/Akt signalling axis, the activated AKT phosphorylates multiple targets and is involved

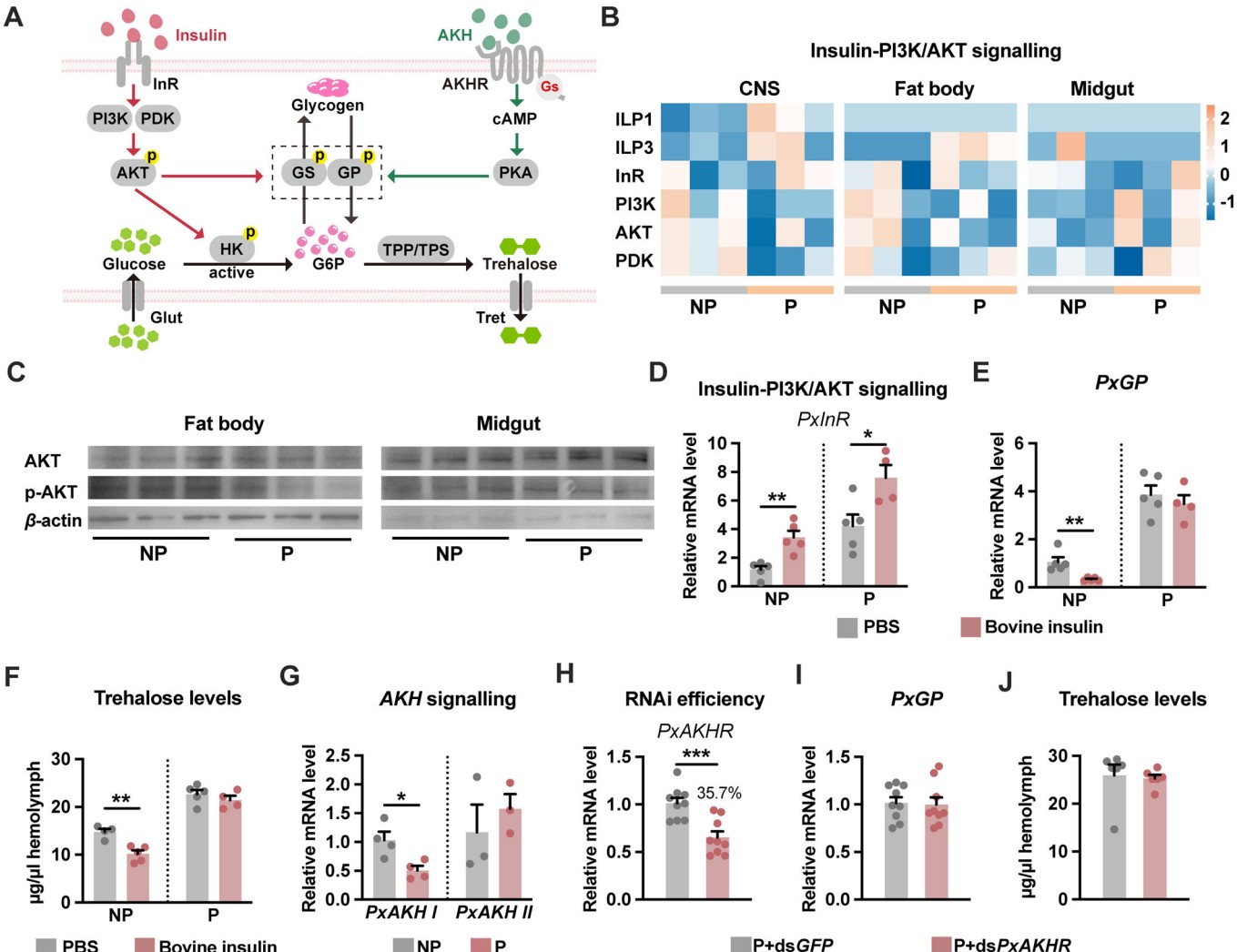

**Figure 3. Trehalose levels are not affected by insulin and AKH signalling after parasitism.**

(A) Schematic diagram of the regulation of glycogenolysis and glycolysis by insulin and AKH signalling. (B) The expression level of insulin-PI3K/AKT signalling-related genes in the CNS, fat body and midgut of 3L instar hosts. Orange and blue colours represent high and low expression levels based on the lgFPKM values, respectively. ILP1 insulin-like peptide 1, ILP3 insulin-like peptide 3, InR insulin receptor, PI3K phosphatidylinositol 3-kinase, AKT serine/threonine protein kinase Akt, PDK 3-phosphoinositide-dependent protein kinase. GenBank Accession Numbers are listed in Appendix Table S2. (C) Immunoblot analysis of total AKT and p-AKT (40 μg/input) in the fat body and midgut. β-actin (10 μg /input) was used as a loading control. (D–F) Whole-body relative *PxInR* (D) and *PxGP* (E) mRNA expression, and haemolymph trehalose concentration (F) from 4M instar hosts treated with bovine insulin (*n* = 4–5). (G) Relative mRNA levels of *PxAKH I* and *PxAKH II* in 3L instar hosts (whole body, *n* = 3–4). (H) RNAi efficiency of ds*PxAKHR* in parasitised host larvae (whole body) at 3 L instar (*n* = 9). ds*PxGFP* was used as the control. (I, J) Relative mRNA levels of *PxGP* from fat body at 3L instar (I, *n* = 9) and haemolymph trehalose concentration at 4M instar (J, *n* = 6) after *PxAKHR* knockdown. Data are represented as mean ± SEM. Two-tailed unpaired Student's *t* test. *P< 0.05, **P< 0.01, ***P< 0.005, not shown P > 0.05. The exact P values are provided in Appendix Table S5. Source data are available online for this figure.

in metabolic homeostasis (Hopkins et al, 2020). Immunoblot analysis further revealed that the levels of total AKT and phosphorylated AKT were comparable between the fat body and midgut of parasitised and non-parasitised larvae (Fig. 3C; Appendix Fig. S3A). Previous reports have shown that the exogenous addition of bovine insulin reduces haemolymph trehalose levels in insects (Kim and Hong, 2015). We next asked whether bovine insulin could correct parasitism-induced hypertrehalosemia. The expression level of *PxInR* was upregulated in both non-parasitised and parasitised hosts by bovine insulin treatment (Fig. 3D). However, haemolymph trehalose levels and GP expression levels were not reduced by bovine insulin in the parasitised host as in the non-

parasitised group (Fig. 3E,F; Appendix Fig. S3B). Thus, hyper-trehalosemia in parasitised hosts cannot be rescued by forced exogenous insulin supplementation, which is a typical phenotype of insulin resistance.

There are two AKH genes in *P. xylostella*, *PxAKH I* and *PxAKH II*. Previous transcriptome analyses revealed that the transcription level of *AKH I* was downregulated in the host after parasitisation (Shi et al, 2015). Our qPCR results also confirmed that the transcription level of *PxAKH I* in the 3 L larvae was decreased after parasitisation, whereas that of *PxAKH II* was not affected (Fig. 3G). The transcription level of *PxGP* in the fat body was unaffected after the elimination of AKH signalling in parasitised *P. xylostella* with

RNAi-based knockdown of the *AKH receptor* (*AKHR*) (Fig. 3H,I). Furthermore, no substantial variations were detected in the amounts of trehalose in the haemolymph or glycogen in the fat body (Fig. 3J; Appendix Fig. S3C,D), suggesting that AKH might not be a cause of parasitism-induced hypertrehalosemia. Collectively, our results demonstrate that although parasitism induces insulin resistance in the host, the consequent hypertrehalosemia is not mediated by changes in insulin or AKH signalling, implying the involvement of additional regulatory mechanisms.

## CvsNPF from teratocytes is responsible for PxGP-mediated glycogenolysis

During *C. vestalis* parasitism, venom and Coteisa vestalis bracovirus (CvBV) are maternal factors that are injected into *P. xylostella* larvae (host 3M stage) along with the wasp egg, and teratocytes derived from the embryonic membrane are released into the host haemocoel upon egg hatching (host 4E stage) (Wang et al, 2018). Thus, we wondered which factor is responsible for the elevated trehalose concentration. To mimic the parasitism process, we injected venom or CvBV particles into the 3M *P. xylostella* larvae and the teratocyte content into 4E larvae, separately. As a result, we found that the concentration of circulating trehalose was increased by 40% at 3L after the injection of CvBV and by 25% at 4M after the treatment of the teratocyte content, whereas venom injection did not have such effects (Fig. 4A). The injection of CvBV and teratocyte content in non-parasitised larvae increases the transcript level of *PxHK* and *PxGP*, respectively (Fig. 4B,C). Thus, our observations suggest that CvBV and the teratocytes may employ different mechanisms to increase the trehalose level in the host haemolymph.

Previous studies on the *C. vestalis* teratocyte transcriptome identified 461 unigenes with signal peptides, which were predominantly categorised into four functional groups: regulation of host development, regulation of host immunity, nutrient metabolism and cellular structure (Gao et al, 2016). We found that *C. vestalis* teratocytes express a short neuropeptide F homologue, named *CvsNPF* (Appendix Fig. S3E), which is an important modulator of sugar metabolism in *Drosophila* (Lee et al, 2008; Oh et al, 2019). Multiple sequence alignments suggested that the CvsNPF prepropeptide had a conserved C-terminal sequence similar to that of other insects (Appendix Fig. S3F). Only one mature peptide sequence was predicted from the CvsNPF prepropeptide, which was highly similar to sNPF from *P. xylostella*. To investigate whether *CvsNPF* was secreted into the parasitised *P. xylostella* larvae, we produced an antibody specific to the CvsNPF prepropeptide. Immunohistochemistry revealed the localisation of CvsNPF in teratocytes (Fig. 4D). Furthermore, the CvsNPF antibody provoked immunological responses in the haemolymph of the parasitised host but not in that of the non-parasitised host (Fig. 4E). A band was observed at a higher molecular weight than expected, which may be due to post-translational modifications, such as glycosylation. Furthermore, LC/MS analysis of the corresponding band identified multiple fragments of the Cv-sNPF propeptide (Appendix Table S3).

We then injected synthetic CvsNPF (sCvsNPF) to assess the function of CvsNPF in *P. xylostella* larvae. The results showed that sCvsNPF significantly increased the trehalose levels in the haemolymph of *P. xylostella* larvae. The trehalose concentration increased by 40% after injections of 0.005 ng, 0.05 ng, or 0.5 ng sCvsNPF, suggesting that this effect was independent of the dose (Fig. 4F). qPCR analysis further showed that sCvsNPF upregulated the expression of *PxGP* but did not affect the transcript levels of *PxHK* and *PxTPS* (Fig. 4G). Because *PxGP* was primarily expressed in the *P. xylostella* fat body, we also detected a considerable decrease in the stored glycogen level in the fat body following sCvsNPF injection (Fig. 4H,I), consistent with the findings in the parasitised hosts (Fig. 1F,G). Thus, we concluded that CvsNPF produced by *C. vestalis* teratocytes promoted glycogenolysis by upregulating *PxGP* expression in the fat body, which led to the accumulation of circulating trehalose.

## CvBV-induced PxsNPF results in high trehalose levels via PxHK-mediated glycolysis

Because sNPF from *C. vestalis* is involved in regulating trehalose metabolism, we wondered whether sNPF from the host *P. xylostella* would have a similar effect on circulating trehalose. Thus, we cloned the full-length cDNA of the *P. xylostella* short neuropeptide F, *PxsNPF*, which has an open reading frame of 543 bp. The amino acid sequence of predicted PxsNPF showed three amidation signals flanked by monobasic or dibasic cleavage sites, suggesting that PxsNPF was cleaved into three putative PxsNPF peptides, PxsNPF-1, PxsNPF-2 and PxsNPF-3. Sequence alignment analysis revealed that the putative PxsNPF-1, PxsNPF-2 and PxsNPF-3 had a conserved C-terminal sequence similar to that of other insects (Appendix Fig. S3F).

We analysed the mRNA expression profiles of *PxsNPF* in the parasitised host larvae. The results showed that the mRNA levels of *PxsNPF* were significantly upregulated in the parasitised larvae compared with those in the non-parasitised larvae at all four stages (3L, 4E, 4M and 4L) (Fig. 5A). To investigate the function of PxsNPF in *P. xylostella*, we separately injected the three synthetic PxsNPFs (sPxsNPFs) into the 3M larvae and detected the trehalose level of host larvae (4M). The results showed that a high dosage (0.5 ng per larva) of sPxsNPF-1 increased the haemolymph trehalose level, whereas sPxsNPF-2 and sPxsNPF-3 did not have this effect (Fig. 5B), suggesting that the trehalose-increasing effect of PxsNPF-1 was dependent on the concentration.

Next, we examined the source of PxsNPF signalling involved in the regulation of systemic trehalose homeostasis. We found that *PxsNPF* mRNA was primarily expressed in the CNS and midgut, and the transcript levels were significantly increased in these tissues after parasitism (Appendix Fig. S4A). Immunofluorescence assays revealed that PxsNPF is localised to the brain of *P. xylostella* and that parasitism led to a massive increase in PxsNPF peptide production (Fig. 5C,D). Strikingly, we found that midgut-derived PxsNPF was expressed by tracheal cells instead of intestinal cells (Appendix Fig. S4B). The PxsNPF-producing tracheal cells surrounded the anterior, middle, and posterior regions of the midgut (Appendix Fig. S4C). Parasitism significantly increased the PxsNPF signal in terminal trachea cells (Fig. 5E,F).

Further analysis revealed that the transcription of *PxHK* was upregulated after sPxsNPF-1 injection, whereas that of *PxGP* and *PxTPS* remained unaffected (Fig. 5G). Moreover, the ds*PxsNPF* injection significantly decreased the expression of *PxsNPF* in parasitised hosts compared with the negative control (Fig. 5H). The *PxsNPF*-knockdown hosts showed decreased trehalose levels in the

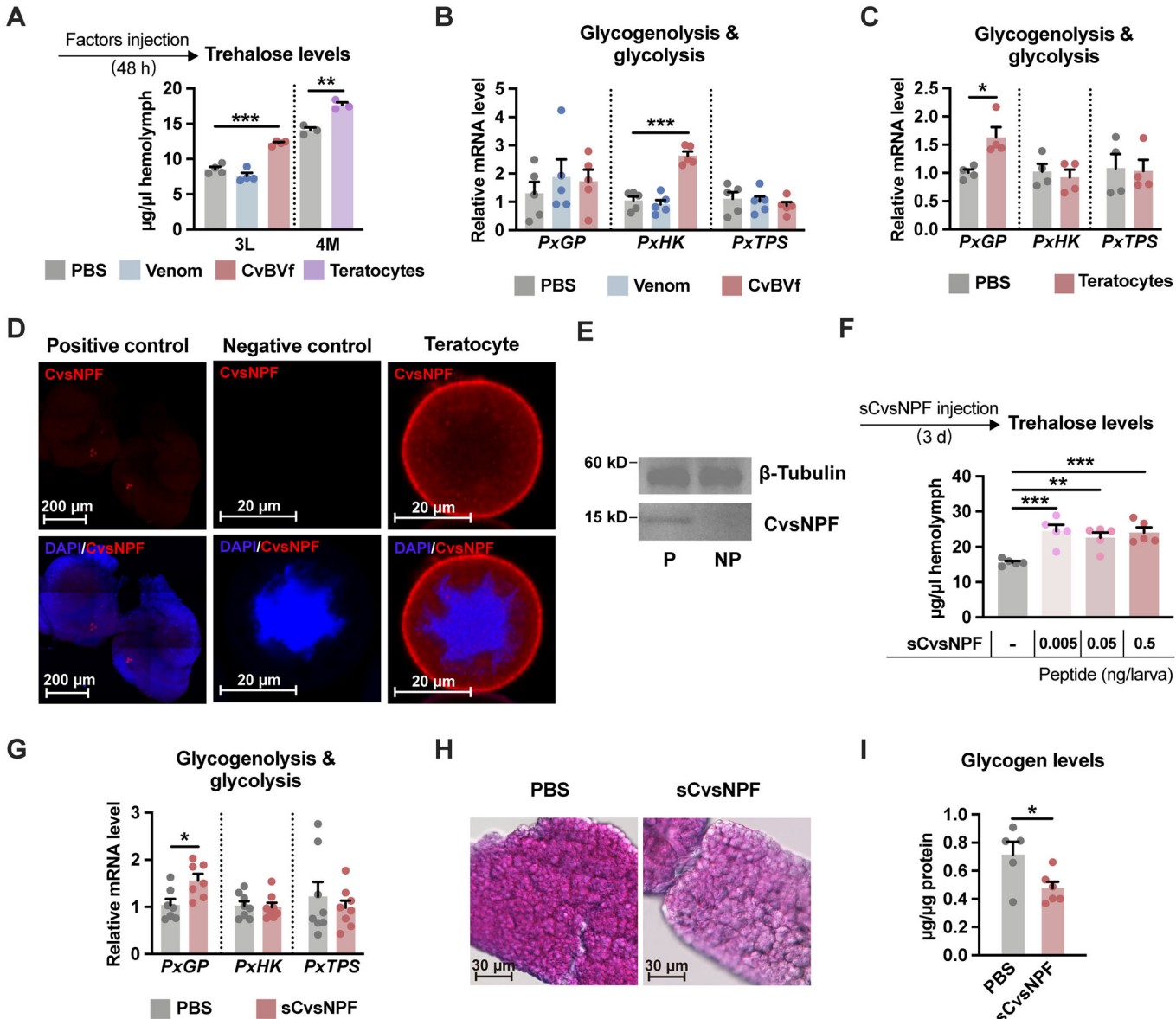

**Figure 4. CvsNPF from teratocytes contributes to the *PxGP*-mediated glycogenolysis.**

(**A**) Effects of venom, CvBV or teratocytes injection on circulating trehalose of *P. xylostella* (*n* = 3–4). (**B, C**) Relative mRNA levels of *PxGP*, *PxHK* and *PxTPS* from *P. xylostella* larvae (whole body) injected with venom, CvBVat 3 L instar (**B**, *n* = 5) or teratocytes at 4M instar (**C**, *n* = 4), respectively. (**D**) Immunostaining of CvsNPF prepropeptide (red) in 3-day-old teratocytes. Positive control shows the immunostaining of CvsNPF neurosecretory cells in the brain of late 2nd instar *C. vestalis* larvae. The negative control shows the results of teratocytes probed without the primary antibody. Nuclei were labelled by DAPI (blue). (**E**) Immunoblot for CvsNPF in parasitised host haemolymph (anti-CvsNPF antibody). β-tubulin was used as a control. (**F**) Circulating trehalose levels after sCvsNPF injection (*n* = 5). (**G–I**) Whole body relative *PxGP*, *PxHK* and *PxTPS* mRNA levels (**G**, *n* = 7–8), PAS staining of the fat body (**H**) and its glycogen content at 4L instar (**I**, *n* = 5–6) from sCvsNPF-treated (0.5 ng) *P. xylostella* larvae. Scale bars, 30 μm. Data are represented as mean ± SEM. *$P< 0.05$, **$P< 0.01$, ***$P<0.005$; ns no significance. The exact *P* values are provided in Appendix Table S5. (**A, B, F**) One-way ANOVA and Tukey's multiple comparison tests; (**C, G, I**) two-tailed unpaired Student's *t* test. Source data are available online for this figure.

haemolymph along with downregulation of *PxHK* expression, which is primarily expressed in the *P. xylostella* midgut (Fig. 5I,J). In contrast, the transcript levels of *PxGP* and *PxTPS* were not affected by ds*PxsNPF* injection (Fig. 5J). Thus, our results suggest that PxsNPF probably promotes trehalose synthesis by increasing the *PxHK* transcript levels in the midgut.

To identify which parasitic factor was responsible for the upregulation of *PxsNPF*, we injected *C. vestalis* venom, CvBV and

teratocytes into the *P. xylostella* larvae separately. We detected an increase in *PxsNPF* expression 24 h after CvBV injection (Fig. 5K), which explained our previous finding that CvBV is responsible for the increase in the trehalose content in parasitised host larvae, and the increase of *PxHK* by CvBV corresponds to the upregulation of PxsNPF-1 (Fig. 4A,B). Overall, the results indicate that CvBV increases the host trehalose levels in the host by increasing the *PxsNPF* transcript levels.

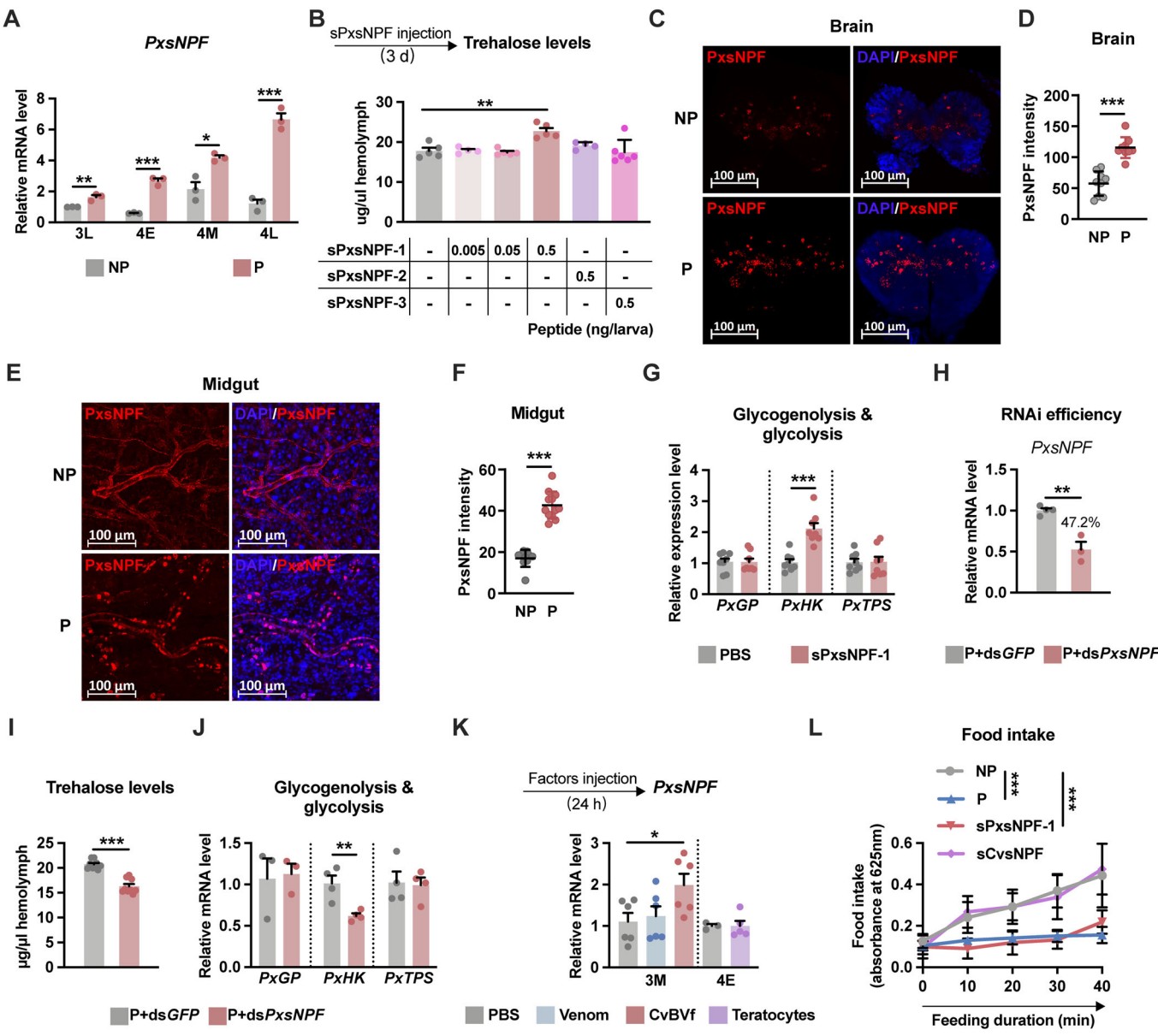

**Figure 5. CvBV-induced PxsNPF results in hypertrhalosmia via *PxHK*-mediated glycolysis.**

(A) Relative mRNA levels of *PxsNPF* (whole body, $n = 3$). (B) Circulating trehalose levels after injections of sPxsNPF-1, sPxsNPF-2 or sPxsNPF-3 ($n = 4$–6).
(C and D) Immunostaining (C) and fluorescent intensity (D) of PxsNPF (red) in the brain of non-parasitised and parasitised *P. xylostella* larvae at 3L instar ($n = 9$).
(E, F) Immunostaining and fluorescent intensity of PxsNPF (red) in the midgut of non-parasitised and parasitised *P. xylostella* larvae at 3L instar, quantified in (F) ($n = 10$).
Nuclei were labelled by DAPI (blue). Magnified views (E) were generated from the areas in the white dashed box of Appendix Fig. S4C. Scale bar, 100 μm. (G) Relative mRNA levels of *PxGP*, *PxHK* and *PxTPS* ($n = 8$) from sPxsNPF-1-treated (0.5 ng) *P. xylostella* larvae (whole body) at 4L instar. (H) RNAi efficiency of ds*PxsNPF* in parasitised host larvae at 4M instar (whole body, $n = 3$–4). ds*PxGFP* was used as the control. (I and J) Circulating trehalose concentration (I, $n = 9$) and relative mRNA levels of *PxGP*, *PxHK* and *PxTPS* (J, $n = 3$–4) from *C. vestalis* parasitised host larvae (whole body) treated with ds*PxGFP* or ds*PxsNPF*. (K) Relative mRNA levels of *PxsNPF* from *P. xylostella* larvae (whole body) injected with venom, CvBV or teratocytes, respectively ($n = 3$–6). (L) Colourimetric quantification of food consumption in non-parasitised, parasitised, and sPxsNPF-1-injected and sCvsNPF-injected *P. xylostella* larvae at the middle 4th instar in a 40-min feeding assay ($n = 10$). Each synthetic peptide was injected at a dose of 0.5 ng per larva. Data are represented as mean ± SEM. *$P<0.05$, **$P<0.01$, ***$P<0.005$, not shown $P > 0.05$. The exact $P$ values are provided in Appendix Table S5. **A, D, F-J**, Two-tailed unpaired Student's *t* test; (**B**, **K**) one-way ANOVA and Tukey's multiple comparison tests; (**L**) two-way ANOVA and Tukey's multiple comparison tests. Source data are available online for this figure.

## PxsNPF inhibits feeding behaviour in host larvae

The intricate involvement of sNPF in feeding and metabolism has been well documented in many insects, and yet, depending on the species, sNPF can act as either a stimulatory or an inhibitory factor (Fadda et al, 2019). We wondered whether such consistency for the changes in the carbohydrate level is dependent on the feeding regulation. The food intake of the parasitised hosts decreased

significantly with increases in the feeding interval (Fig. 5L; Appendix Fig. S4D). Consistently, we observed a similar inhibitory effect on feeding after injection of sPxsNPF-1 but not sCvsNPF (Fig. 5L; Appendix Fig. S4D). Compared with that of the control group, the feed interval time was likewise longer after the injection of sPxsNPF-1, but the feed time was unaffected. These findings imply that PxsNPF exhibits an inhibitory role in the control of feeding and hypertrehalosemia caused by parasitoid wasps may not depend on feeding behaviour regulation.

## CvsNPF and PxsNPF activate the PxsNPF receptor via distinct G protein-mediated pathways

Neuropeptides exert their biological functions by binding to specific membrane receptors, most of which are G protein-coupled receptors (GPCRs) (Cui and Zhao, 2020). Bioinformatics and genome analysis revealed that *P. xylostella* possesses a single sNPF receptor homologue, *PxsNPFR*. Sequence and phylogenetic analyses confirmed that *PxsNPFR* is a member of the GPCR family (Appendix Fig. S5A,B). In *P. xylostella*, *PxsNPFR* was most highly expressed in the midgut, followed by the fat body and the CNS, which were CvsNPF/PxsNPF-targeted tissues (Appendix Fig. S6A). Immunofluorescence results further confirmed that PxsNPFR in the midgut was localised in both terminal tracheal cells and intestinal cells (Appendix Fig. S6B). By combining the *PxsNPFR* interference and sNPF treatment (Fig. 6A–C), we found that sCvsNPF-induced *PxGP* expression and sPxsNPF-1-induced *PxHK* expression were abolished following ds*PxsNPFR* treatment in non-parasitised host larvae, suggesting that both CvsNPF and PxsNPF exert their effects through PxsNPFR. Similar to the RNAi-mediated knockdown of sNPF, knockdown of sNPFR in parasitised host larvae also led to a significant reduction in trehalose levels (Appendix Fig. S6C,D). Despite the unchanged pupation rate and eclosion duration, other parameters—including the eclosion, the fecundity and longevity of male wasps fed with water—were comparable between wasps emerging from ds*sNPFR*-treated hosts and those from ds*GP*-treated hosts (Appendix Fig. S6E–K). These results suggest that the sNPF/sNPFR signalling facilitates the larval development of parasitic wasps by modulating host trehalose levels.

To examine the PxsNPFR-mediated signalling, we constructed a pcDNA 3.1$^{(+)}$/*PxsNPFR* plasmid and transfected it into HEK293 cells. The staining results showed that the PxsNPFR signal colocalised with the cell membrane (Appendix Fig. S7A,B). We next assayed cAMP production and $Ca^{2+}$ mobilisation in the cells to confirm whether the cloned PxsNPFR was the functional receptor for sPxsNPF-1 and sCvsNPF. As revealed in Fig. 6D, only the sCvsNPF inhibited forskolin-stimulated cAMP accumulation in a concentration-dependent manner with an $EC_{50}$ of ~0.402 μM, whereas an inhibitory effect was not observed in the transfected cells treated with sPxsNPF. Moreover, both sPxsNPF-1 and sCvsNPF elicited a rapid and transient increase in intracellular $Ca^{2+}$ mobilisation (Fig. 6E,F). Interestingly, the sCvsNPF showed stronger activity in evoking intracellular $Ca^{2+}$, suggesting that PxsNPFR exhibits a stronger affinity for CvsNPF than for PxsNPF. These results suggest that the CvsNPF-PxsNPFR complex interacts with the $G_{i/o}$ protein, leading to a clear inhibition of cAMP accumulation and that the PxsNPF-PxsNPFR interaction regulates $G_q$ protein-mediated signalling.

To further investigate the pathways downstream of PxsNPF and CvsNPF activation, we separated the fat body and midgut from the host and stimulated the tissue with synthetic peptides and specific inhibitors of $G_q$ and $G_i$ signalling ex vivo. The inhibition of PLCβ, the key downstream effector of $G_q$, by treatment with U73122, abolished the facilitatory effect of sPxsNPF-1 on *PxHK* (Fig. 6G). The incubation of the fat body with pertussis toxin (PTX), which prevents receptor coupling to $G_{i/o}$, decreased the sCvsNPF activation of *PxGP* (Fig. 6H), indicating that PxsNPFR selectively stimulates $G_i$ protein and $G_q$ protein coupling.

To investigate signalling signatures associated with PxsNPF or CvsNPF, we performed a transcriptome analysis of the *P. xylostella* fat body and midgut following injection of sPxsNPF-1 or sCvsNPF. The intersection of these two sets of DEGs revealed that only 3.6% (157 transcripts) were found in both sets, suggesting the existence of significant differences in the downstream transcript profiles modulated by sCvsNPF and sPxsNPF-1 signalling, as corroborated by the KEGG enrichment analysis results (Appendix Fig. S8A–E and Datasets EV1 and 2). Activated $G_i$ proteins subsequently block the cyclic adenosine 3,5-monophosphate (cAMP) response, whereas $G_q$ proteins mobilise calcium and further modulate downstream effectors, such as phospholipase C (PLC) (Ritter and Hall, 2009). We observed that genes in the PKA pathway were downregulated by sCvsNPF treatment (Appendix Fig. S8F). In addition, genes involved in the $G_q$ activation, such as PLC and inositol-1,4,5-trisphosphate ($IP_3$), were significantly upregulated after sPxsNPF-1 injection (Appendix Fig. S8G), thereby stimulating the release of $Ca^{2+}$ from the endoplasmic reticulum (Seyedabadi et al, 2019). This finding is consistent with the observed increase in the intracellular calcium levels in HEK293/PxsNPFR cells after sPxsNPF-1 stimulation (Fig. 6E). Collectively, these RNA-seq data further illustrate the ability of PxsNPFR to regulate sugar homeostasis by signalling through multiple G protein pathways.

## Molecular docking predicts the interaction of CvsNPF and PxsNPF-1 with PxsNPFR

To compare the critical interactions of the CvsNPF-PxsNPFR and PxsNPF1-PxsNPFR complexes, we performed molecular modelling and docking based on AlphaFold2-predicted protein structures. Comparison of the two peptide-bound receptor complex structures revealed distinct conformational changes in the extracellular region, TM core and intracellular region of the receptor domains. The extracellular tip of TM1 in the CvsNPF-bound receptor structure is moved outwards by 1.2 Å compared with that in the PxsNPF-bound structure (Fig. 6I). Moreover, the α-helix on extracellular loop (ECL) 3 of the PxsNPF-1-bound receptor is not found in the structure of the CvsNPF-bound receptor. In the intracellular region, the intracellular loop (ICL) 3 of the CvsNPF-bound receptor constricts inwards, which plays a key role in the selectivity of GPCR for G protein coupling (Seyedabadi et al, 2019). The C-terminal sequence of sNPF is highly conserved and important for sNPF-bound receptor activation (Nassel and Wegener, 2011). In both the PxsNPF1 and CvsNPF peptide-bound PxsNPFR structures, the C-terminus of the peptide is located at the bottom of the ligand-binding pocket and inserts into a cavity shaped by helices II to VII (Fig. 6I). The side chains of L9, R10 and F11 in CvsNPF overlay well with those in PxsNPF1 (Fig. 6J). Compared with those of the CvsNPF-bound receptor

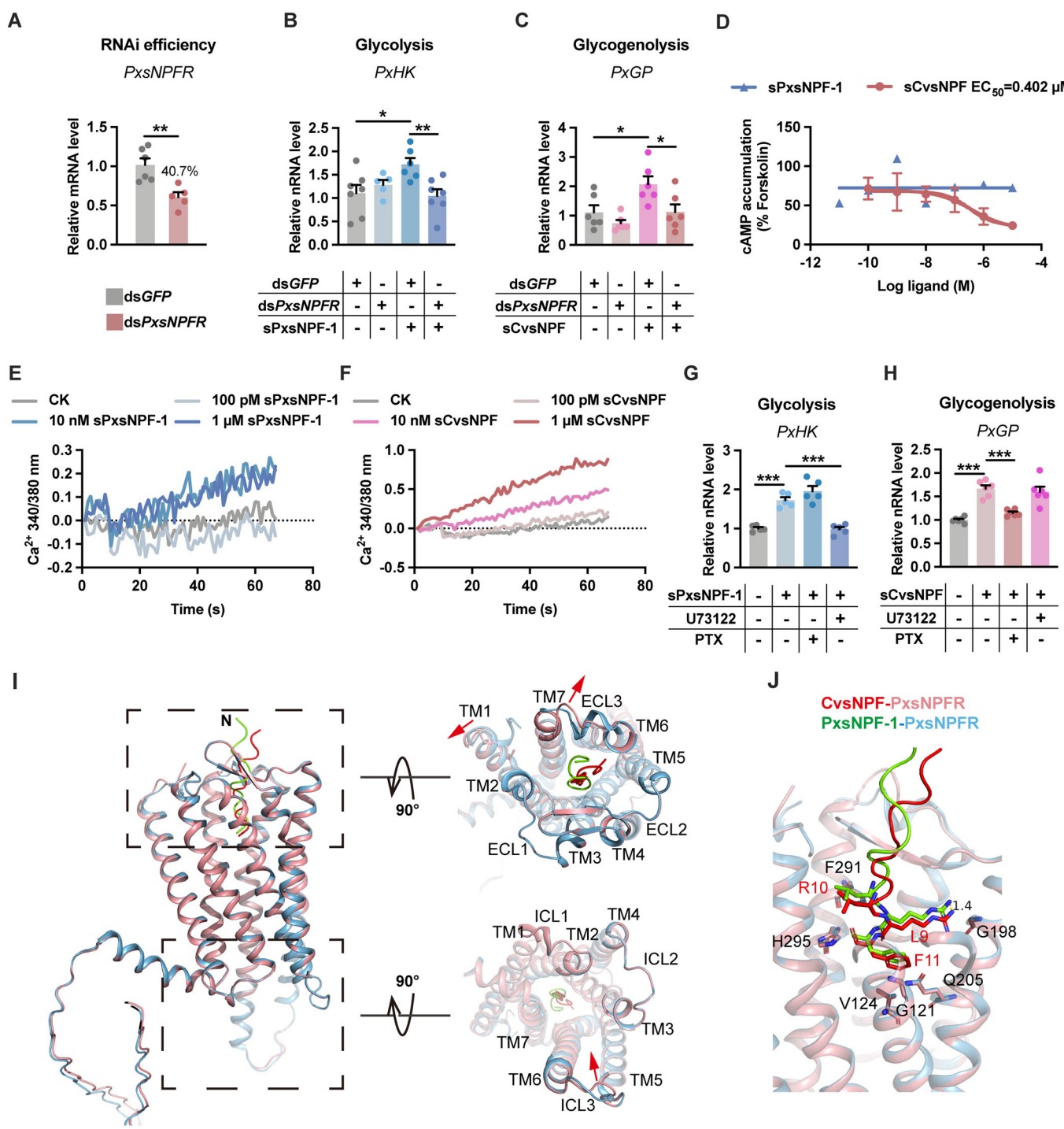

structure, the R10 and F11 residues of PxsNPF1 form three additional hydrogen bonds with residues E186 and T97 in the receptor.

Previous studies have shown that the N-terminal of the NPY plays a role in mediating receptor-peptide recognition (Tang et al, 2022). The amount of buried surface area at the residue reveals the binding affinity to the receptor (Chen et al, 2013). For the N-terminal residues (S1-R3) of CvsNPF, the mean percentage of

the buried surface area was ~57%, and the mean percentage of the buried surface area of the N-terminal residues in PxsNPF-1 was ~67.7%, which revealed that the interaction of the N-terminus of PxsNPF-1 with the receptor was stronger than that of CvsNPF (Appendix Fig. S9). These variations in the sNPF-PxsNPFR binding pocket support the specific binding of PxsNPFR to PxsNPF-1 or CvsNPF, which also partially explains the differences in downstream signalling.

**Figure 6.  CvsNPF and PxsNPF activate the PxsNPF receptor with distinct downstream pathways.**

(A) RNAi efficiency of ds*PxsNPFR* in host larvae (whole body) at 3 L instar (*n* = 5–6). ds*PxGFP* was used as the control. (B, C) Relative mRNA levels of *PxHK* (B) and *PxGP* (C) in non-parasitised host larvae (whole body) at 3 L instar upon synthetic sNPF (0.5 ng) or ds*PxsNPFR* treatment (*n* = 6–7). (D) Dose-dependent inhibition of forskolin-induced cAMP accumulation in PxsNPFR–expressing HEK293 cells in response to sPxsNPF-1 or sCvsNPF. Cells were treated with 10 μM forskolin and various concentrations of the indicated peptide for 15 min at 37 °C (*n* = 3). (E, F) Intracellular $Ca^{2+}$ mobilisation mediated by sPxsNPF-1 (E) and sCvsNPF (F) in PxsNPFR–expressing HEK293 cells. (G, H) Relative mRNA levels of *PxHK* in the midgut (G, *n* = 5) and *PxGP* in the fat body (H, *n* = 6) of host larvae incubated with the indicated sNPF peptide (20 μM) mixed with $G_i$ inhibitor PTX (0.4 μM), PLCβ inhibitor U73122 (10 μM), or DMSO. (I) Structural alignment between CvsNPF (red)-PxsNPFR (pink) and PxsNPF-1 (green)-PxsNPFR (light blue) complex. Views from the extracellular (right-upper panel) and cytoplasmic (right-lower panel) sides are shown on the right. The red arrow represents the movement of the TM helix, extracellular loop (ECL) or intracellular loop (ICL) in the CvsNPF-PxsNPFR complex compared to the PxsNPF-1-bound PxsNPFR. (J) Structure of the hydrophobic region at the bottom of the binding pocket upon CvsNPF and PxsNPF-1 binding. The PxsNPFR residues (pink and light blue), C-terminal tail of CvsNPF (red) and PxsNPF1 (green) that participate in sNPF binding are presented as sticks. Data are represented as mean ± SEM. *$P<0.05$, **$P< 0.01$, ***$P< 0.005$, not shown $P > 0.05$. The exact $P$ values are provided in Appendix Table S5. (A) Two-tailed unpaired Student's *t* test; (B–H) one-way ANOVA and Tukey's multiple comparison tests. Source data are available online for this figure.

## Discussion

Parasites adopt various strategies to manipulate the physiological system of the host to ensure their survival (Pennacchio et al, 2014). In this study, we found that the host haemolymph carbohydrates, especially trehalose, the major circulatory sugar, changed after parasitisation. The increase in host trehalose throughout the parasitism process is crucial for the fitness of wasp offspring after emergence. Importantly, we characterised two sources of sNPF, CvBV-induced PxsNPF and teratocyte-derived CvsNPF, which act as modulators of carbohydrate anabolism and catabolism to increase the circulatory trehalose levels in the *C. vestalis* parasitised hosts.

The development and survival of parasites largely depend on host nutrition. It is well known that metamorphosis requires reprogramming of the carbohydrate metabolism (Nishimura, 2020). Although the carbohydrate compositions have not been fully characterised, our results strongly suggest that low trehalose levels in the host attenuate wasp offspring carbohydrate metabolism and development in wasp offspring (Fig. 2B–D). Similar pupation delays and eclosion deficiencies have been observed in *Drosophila* with low trehalose levels, which led to *Tps1* mutant lethality during the late pupal stage (Matsuda et al, 2015; Nishimura, 2020). Adult parasitoid wasps are especially sensitive to nutrient deprivation, and the lifespan of many parasitoid species significantly decreases in the absence of sugar (Olson et al, 2000). In adult wasps, low glycogen levels are not crucial for physical fitness or lifespan under feeding conditions but are critical under low-carbohydrate conditions. This impact is notably stronger in male offspring than in female offspring. Meanwhile, low glycogen levels in the wasp larvae decreased oviposition ability in mated female wasps—an effect largely attributable to glycogen reduction in male offspring. Indeed, females and males show differences in their mechanisms for allocating nutritional resources to survival and reproduction, and their longevity also differs when exposed to nutritional stress. For instance, female *Pachycrepoideus vindemmiae* are more resistant to water and honey deficiency than males (Da Silva et al, 2019). Our findings suggest that these wasps exhibit greater sensitivity to sugar deficiency, potentially leading to an adverse impact on female reproduction. As an important biocontrol agent, it will be intriguing to explore sexual dimorphism in the nutritional requirements of this wasp to facilitate mass-rearing and application.

Alteration of the host physiology and behaviour by parasites is a widespread phenomenon, and the underlying mechanism is just beginning to be deciphered. It has been suggested that diverse parasites, ranging from viruses to parasitic worms and also some parasitic insects, manipulate their host through the central and peripheral nervous system (Hughes and Libersat, 2018). For instance, the jewel wasp *Ampulex compressa* injects venom directly into the CNS of the cockroach *Periplaneta americana*, achieving a local central paralysis to facilitate subsequent stings (Haspel et al, 2003). The effect of the venom appears to be mediated by two neuromodulators: GABA and dopamine (Hughes and Libersat, 2018). In some host-parasitoid wasp systems, host neuroendocrine alterations are generally caused by three powerful weapons (venom, PDV and teratocyte) of parasitoid wasps (Hughes and Libersat, 2018; Shi et al, 2015). During parasitism, teratocytes, which are larger in size and exhibit increased ploidy but barely divide, express a cocktail of genes and exhibit multiple functions to regulate host homeostasis (Gao et al, 2016; Wu et al, 2023). In this case, we discovered that sNPF secreted by teratocytes act as a humoral factor to modulate the host physiology. The concept of molecular mimicry, in which parasites share certain antigens with their hosts, has been known for a long time (Inal, 2004). The highly conserved CvsNPF expressed by teratocytes is like a neurohormone mimicry of PxsNPF, which targets distinct tissues but exhibits comparable biological functions, representing novel evidence of hormone crosstalk in host–parasite relationships.

CvBV circles are integrated into the genome of *P. xylostella* and express hundreds of functional genes, which may be involved in host neuroendocrine changes (Shi et al, 2015; Wang et al, 2021). Although our study does not yet provide direct mechanistic evidence, previous findings demonstrated that insect tracheal epithelial cells serve as a major conduit for systemic viral dissemination (Engelhard et al, 1994). The enlarged tracheal cell nuclei and the upregulation of sNPF in these cells collectively suggest that they are a direct target tissue of CvBV. It is well established that tracheal and intestinal tissues communicate extensively (Perochon et al, 2021; Tamamouna et al, 2021). Terminal tracheal cells play a role in regulating the metabolic state of gut epithelial cells by mediating both systemic and local neuronal signals (Linneweber et al, 2014). It would be interesting to investigate how tracheal sNPF/sNPFR signalling regulates gut metabolism and the mechanism by which CvBV gene expression increases sNPF levels.

The insulin/ILP function in metabolism, growth, reproduction, and ageing has been thoroughly investigated due to its evolutionarily conserved function in the animal kingdom. In normal *P. xylostella* larvae, insulin treatment reduces the blood trehalose level, possibly through preventing the breakdown of glycogen as it does in mammals and other insects (Broughton et al, 2008; Sato-Miyata et al, 2014). Unexpectedly, we observed insulin resistance in parasitised hosts. Except for the activation of InR, the cascade of downstream signalling molecules (AKT and GP) appears to be unaffected. Similar results were found in the AKH-AKHR pathway. These results support the notion that systemic metabolism modulation under parasitism conditions is distinct from that under homoeostatic conditions (Medzhitov, 2021). Given the complex roles of the three parasitic factors, we propose that certain factors from the parasitoid wasp may block signal transduction of these two pathways.

sNPF is a multifunctional hormone involved in the regulation of various physiological processes in insects, such as feeding, moulting, courtship, development and so on (Cholewinski et al, 2024). Depending on the species, sNPF either promotes feeding or inhibits food intake. In *D. melanogaster*, overexpression of *sNPF* increases food intake in larvae and adults, resulting in obesity or larger body size (Lee et al, 2004). Conversely, feeding inhibition by sNPF is also observed in mosquitoes and locusts (Christ et al, 2018; Dillen et al, 2014; Liesch et al, 2013). Though loss-of-function to detect the role of sNPF in *P. xylostella* larvae is absent, the consistent inhibitory effect on food uptake by parasitism, which induces the expression of PxsNPF, and injection of PxsNPF-1, suggests that PxsNPF exerts a negative role in food uptake, while CvsNPF does not affect host feeding. It is well known that blood sugar level usually positively correlates with feeding behaviour (Nagata, 2019). Based on our findings, we propose that parasitism-induced hypertrehalosemia may not be driven by PxsNPF-mediated feeding behaviour regulation.

Besides feeding regulation, sNPF interacts with insulin/ILP signalling to regulate energy metabolism and growth (Hong et al, 2012; Kapan et al, 2012; Lee et al, 2008). In *D. melanogaster*, the circulating levels of glucose and trehalose in the haemolymph are elevated in *sNPF* mutant *Drosophila* due to suppressed insulin signalling (Kapan et al, 2012; Lee et al, 2008). A recent study revealed that CN neurons expressing sNPF reciprocally regulate the insulin and AKH to maintain carbohydrate homeostasis, i.e., sNPF activates insulin expression in insulin-producing cells but inhibits AKH activity in the corpus cardiacum (Oh et al, 2019). sNPF therefore seems to function as a signal-integration hub by regulating insulin and AKH release. As the transcript data show, the upregulation of ILP1 and the downregulation of AKH I in the CNS of parasitised larvae suggest a possible interplay between PxsNPF and ILP or AKH. However, results regarding the downstream signalling molecules in target tissues reveal that ILP and AKH are not major contributors, suggesting a direct manipulation of sugar homeostasis by sNPF. A few reports indicate that blocking peripheral Y1 receptor (Y1R) signalling improves insulin secretion from pancreatic $\beta$-cells and enhances insulin-stimulated glucose uptake in skeletal muscle (Yang et al, 2022). Selective peripheral antagonism of Y1R leads to measurable improvements in glucose tolerance under conditions of diet-induced obesity (Yan et al, 2021). These findings indicate that sNPF/NPY signalling is likely to be an alternative pathway involved in sugar homeostasis regulation.

Most neuropeptides elicit their biological effects through the GPCRs that are present on the cell surface (Cui and Zhao, 2020). GPCRs activated by ligands interact with different effector proteins, including $G_q$, $G_s$ and $G_i$, triggering intracellular signalling pathways (Ritter and Hall, 2009; Seyedabadi et al, 2019). The interaction between sNPF and its receptor in different cells leads to opposing outcomes in the *Drosophila* brain due to the different G protein signalling pathways coupled by sNPFR (Oh et al, 2019). In our studies, the difference in cAMP or $Ca^{2+}$ in HEK293/PxsNPFR cells after treatment with CvsNPF or PxsNPF suggested that CvsNPF has a stronger affinity for PxsNPFR than for PxsNPF, and the effects of the two sNPF may be mediated by $G_i$ proteins in the fat body and by $G_q$ protein in the midgut, consistent with the results of the inhibitor injection experiments (Fig. 6G,H). Likewise, the inhibition of cAMP/PKA cascade is much more like peripheral NPY-Y1R interaction, which plays a major role in the respiratory system, physiology and endocrine system (Shende and Desai, 2020). This functional specificity is largely supported by structural prediction and receptor docking analysis (Fig. 6I). Regarding the differential effects of sPxsNPFs (sPxsNPF-1, -2 and -3) on increasing circulating trehalose level or reducing food intake, the variation may be attributed to differences in the binding affinities between the ligands and their respective receptors. A similar phenomenon has been observed in *Bombyx mori*. Despite the co-expression of multiple sNPF isoforms, only sNPF-2 significantly enhanced feeding behaviour, while sNPF-1 and sNPF-3 were ineffective (Matsumoto et al, 2019). Despite the high similarity of amino acid sequences between PxsNPF and CvsNPF, the structural difference in the sNPF-PxsNPFR binding pocket further supports the specific binding of PxsNPFR to PxsNPF or CvsNPF.

Disorders of carbohydrate metabolism are common features in parasitology. A few studies report examples of parasite-induced host carbohydrate metabolism regulation, which has been suggested that the flow of carbohydrates is usually undirected and mediated by multiple factors, implying that the exact regulatory mechanism may be much more sophisticated in parasitised organisms (Ramos et al, 2022; Shah-Simpson et al, 2017; Vandermosten et al, 2018). For example, infection with malaria and other protozoan parasites often leads to hypoglycaemia, and the survival and reproduction of these intracellular pathogens are usually sensitive to exogenous glucose (Ramos et al, 2022; Shah-Simpson et al, 2017). When the blood glucose level decreases to below a threshold that would lead to the death of the host, adrenal hormones prevent severe hypoglycaemia by exhausting hepatic glycogen (Vandermosten et al, 2018). Intriguingly, the regulation of glycaemia during malaria infection is independent of insulin. Combining our findings, these results support the notion that metabolism modulation under parasitism conditions is distinct from that under homeostatic conditions (Medzhitov, 2021).

In conclusion, by characterising the roles of two sNPF homologues in the parasitoid wasp-host system, our study discovers a cross-species hormone regulation of the host's metabolism, thereby addressing a crucial gap in the comprehension of how parasites modulate host carbohydrate metabolism. Once parasitism occurs, CvBV-induced *PxsNPF*, which is mainly expressed in the *P. xylostella* brain and midgut tracheal, promotes glycolysis through PxsNPF- PxsNPFR signalling in the midgut. In addition to the *PxsNPF*, parasitoid wasp teratocyte-derived CvsNPF has an even stronger affinity to PxsNPFR and activates a distinct

downstream pathway that reprograms carbohydrate metabolism in the host through GP-mediated glycogenesis in the fat body. Our findings provide a paradigm of molecular mimicry by demonstrating sNPF-sNPFR coevolution as a driver of parasitic metabolic hijacking and thus reveal the complexity of metabolism modulation in the host–parasite interplay.

# Methods

### Reagents and tools table

| Reagent/resource | Reference or source | Identifier or catalogue number |
| --- | --- | --- |
| **Experimental models** | | |
| *Cotesia vestalis* | Wang et al, 2018 | N/A |
| *Plutella xylostella* | Wang et al, 2018 | RRID: NCBITaxon_51655 |
| Human embryonic kidney 293 (HEK293T) | Cellosaurus | RRID: CVCL_0063 |
| **Recombinant DNA** | | |
| Plasmid: pcDNA 3.1(+)/ PxsNPFR | This paper | N/A |
| **Antibodies** | | |
| Mouse anti-β-Actin | ABclonal | AC004 |
| HRP Goat Anti-Rabbit IgG | ABclonal | AS014; |
| HRP Goat Anti-Mouse IgG | ABclonal | AS003; |
| Rabbit anti-phospho-AKT (Ser473) | Abmart | Cat# T0067 |
| Rabbit anti-AKT | Huabio | Cat# ET1609-51 |
| Alexa Fluor 488 | Invitrogen | A32731 |
| Alexa Fluor 594 | Invitrogen | A32740 |
| Rabbit polyclonal anti-CvsNPF prepropeptide: AENYLDYGEENADRNI | This paper | N/A |
| Rabbit polyclonal anti-PxsNPF prepropeptide: QALSQYDSVAQSAQEAA | This paper | N/A |
| Rabbit polyclonal anti-PxsNPFR (1-42 AA) | This paper | N/A |
| **Oligonucleotides and other sequence-based reagents** | | |
| Primers | This paper | See Appendix Table S4 |
| **Chemicals, enzymes and other reagents** | | |
| Pertussis toxin (PTX) | Tocris Bioscience | 3097 |
| U73122 | MedChemExpress | HY-13419 |
| Dulbecco's PBS (D-PBS) | Sigma-Aldrich | D8537 |
| TNM-FH medium | HyClone | SH30280.03 |
| RNA-easy Isolation Reagent | Vazyme | R701-01 |
| KOD OneTM PCR master Mix | TOYOBO | KMM-201 |
| RNA isolater Total RNA Extraction Reagent | Vazyme | R401-01 |

| Reagent/resource | Reference or source | Identifier or catalogue number |
| --- | --- | --- |
| THUNDERBIRD® SYBR qPCR Mix | Toyobo | QPS-201 |
| Gold Antifade Mountant | Invitrogen | S36937 |
| Forskolin | Beyotime | S1612 |
| IBMX | Sigma-Aldrich | I7018 |
| Fura-2/AM | Dojindo | Cat# 108964-32-5 |
| Bradford Reagent | Thermo Fisher | Cat# 23200 |
| Amyloglucosidase | Sigma-Aldrich | A7420 |
| Blue food dye | Sigma | 861146 |
| sCvsNPF: SQRSPSLRLRFamide | This paper | N/A |
| sPxsNPF-1: SVRSPSRRLRFamide | This paper | N/A |
| sPxsNPF-2: DTRQPVRLRFamide | This paper | N/A |
| sPxsNPF-3: SVRAPSMRLRFamide | This paper | N/A |
| Carbohydrates Kit | Sigma-Aldrich | CAR10 |
| D-(+)-Trehalose dihydrate | Sigma-Aldrich | T5251; CAS: 6138-23-4 |
| D-Sorbitol | Sigma-Aldrich | 85529; CAS: 50-70-4 |
| DMSO | Sigma-Aldrich | D2650 |
| **Software** | | |
| DNASTAR 5.02 | https://www.dnastar.com/software/ | N/A |
| ZEN 2.3 imaging software | https://www.zeiss.com/microscopy/en/products/software/zeiss-zen.html | RRID:SCR_013672 |
| GraphPad Prism 9 | https://www.graphpad.com/features | RRID:SCR_002798 |
| PRED COUPLE | http://bioinformatics.biol.uoa.gr/PRED-COUPLE | RRID:SCR_006193 |
| CCTOP | https://cctop.ttk.hu/ | RRID:SCR_016963 |
| Multi-state modelling of the GPCR and kinase using AlphaFold2 | https://github.com/huhlim/alphafold-multistate | N/A |
| SIGNALP 4.0 | http://www.cbs.dtu.dk/services/SignalP/ | RRID:SCR_015644 |
| AlphaFold2-Multimer | https://github.com/sokrypton/ColabFold (Evans et al, 2022) | N/A |
| SnapGene 6.0.2 | www.snapgene.com | RRID:SCR_015052 |
| Ligplot 2.2.8 | https://www.ebi.ac.uk/thornton-srv/software/LigPlus/download.html (Laskowski and Swindells, 2011) | RRID:SCR_018249 |
| PyMOL 2.1 | http://www.pymol.org/ | RRID:SCR_000305 |
| PDBePISA | https://www.ebi.ac.uk/pdbe/pisa/ | RRID:SCR_015749 |

| Reagent/resource | Reference or source | Identifier or catalogue number |
| --- | --- | --- |
| MEGA 11 | https://www.megasoftware.net (Tamura et al, 2021) | RRID:SCR_023017 |
| **Critical commercial assays** | | |
| SuperScript III First-Strand Synthesis System | Invitrogen | 18080051 |
| 5 min TA/Blunt-Zero Cloning Kit | Vazyme | C601 |
| cAMP-Glo™ Assay | Promega | Cat# V1501 |
| Glucose (HK) Assay kit | Sigma-Aldrich | Cat# GAHK20-1KT |
| Glucose (GO) Assay Kit | Sigma-Aldrich | Cat# GAGO20 |
| Glucose-6-Phosphate Assay Kit | Sigma-Aldrich | Cat# MAK014-1KT |
| Glycogen periodic acid schiff (PAS/Hematoxylin) stain kit | Solarbio | G1281 |
| FastPure Gel DNA Extration Mini Kit | Vazyme | DC301 |
| T7 High Yield RNA Transcription Kit | Vazyme | TR101-01 |
| Minute™ Total Protein Extraction Kit for Insects | Invent Biotechnologies | SA-07-IS |
| Minute™ Plasma Membrane/Protein Isolation and Cell Fractionation Kit | Invent Biotechnologies | SM-005 |

## Insect rearing and parasitisation

*C. vestalis* and *P. xylostella* used in this study have been continuously maintained for decades in our lab (Wang et al, 2018), and both insects were reared at 25 °C, 65% relative humidity, and 14 h light: 10 h dark photoperiod. *P. xylostella* larvae were maintained on an artificial diet. *C. vestalis* larvae were bred on *P. xylostella* larvae, and adult wasps were fed with 20% (v/v) honey solution.

For parasitisation experiments, middle 3rd instar *P. xylostella* larvae were individually exposed to a single mated female wasp until oviposition was observed.

## Cell line

The human embryonic kidney cell 293 (HEK293) cells were maintained in DMEM supplemented with 10% foetal bovine serum (Gibco, USA) at 37 °C with 5% $CO_2$. Cell culture medium was added with 1000 units/L penicillin and 1000 mg/L streptomycin.

## Sequence and phylogenetic analysis

The amino acid sequences of CvsNPF, CvsNPFR, PxsNPF and PxsNPFR were obtained from the genomes of *C. vestalis* and *P. xylostella*, respectively (Shi et al, 2019; You et al, 2013). Deduced amino acid sequences of CvsNPF, PxsNPF and PxsNPFR were analysed using DNASTAR (Version 5.02, DNASTAR Inc., USA). Database searches were performed with BLASTP (http://

www.ncbi.nlm.nih.gov/). The signal sequence was predicted by SIGNALP 4.0 (http://www.cbs.dtu.dk/services/SignalP/). Transmembrane segments were predicted using the topology prediction server CCTOP (https://cctop.ttk.hu/). The web server programme PRED COUPLE (http://bioinformatics.biol.uoa.gr/PRED-COUPLE) was used to predict GPCR coupling specificity. The protein sequence alignment was created by SnapGene 6.0.2 software (www.snapgene.com) using the algorithm Clustal Omega. To construct an sNPFR phylogenetic tree, selected protein sequences were aligned by ClustalW. A phylogenetic tree was constructed using the neighbour-joining method, followed by 1000 bootstrap tests in MEGA version 11 (Tamura et al, 2021).

## Three-dimensional structure prediction and molecular docking

Prediction and building of the activation state structure of PxsNPFR was based on methods reported previously (Heo and Feig, 2022). PxsNPF-bound and CvsNPF-bound PxsNPFR complexes were modelled using a modified AlphaFold2-Multimer following the instructions on the website https://github.com/sokrypton/ColabFold (Evans et al, 2022). The predicted activation state of the receptor model was input as a custom template. The 2D and 3D peptide-receptor interaction models were visualised by Ligplot 2.2.8 (Laskowski and Swindells, 2011) and PyMOL 2.1, respectively. The protein interface analyses were performed by PDBePISA (https://www.ebi.ac.uk/pdbe/pisa/).

## cDNAs cloning

Genes of *P. xylostella* were identified by analysing the expressed sequence tags (ESTs) in the DBM transcriptome apparatus and genome (Shi et al, 2015; You et al, 2013). CvsNPF was identified by analysing the ESTs in the transcriptome data of *C. vestalis* teratocytes(Gao et al, 2016). cDNA library was constructed by the SuperScript III First-Strand Synthesis System (Invitrogen) from total RNA of 2nd to 4th instar *P. xylostella* larvae or 1st to 3rd instar *C. vestalis* larvae, which was extracted with the RNA-easy Isolation Reagent (Vazyme) following the manufacturer's instructions. The entire coding regions (CDS) of genes were amplified from the cDNA library by KOD One™ PCR master Mix (Toyobo) with specific primers (Appendix Table S4) and the following conditions: initial denaturation for 5 min at 95 °C, then amplification for 1 min at 95 °C 1 min at 56–60, 1 min at 72 °C for 30 cycles, followed by a 10 min 72 °C incubation. All PCR-amplified DNA products were purified by FastPure Gel DNA Extraction Mini Kit (Vazyme) and inserted into pCE2 TA/Blunt-zero Vector by 5 min TA/Blunt-Zero Cloning Kit (Vazyme) and sequenced by Tsingke Biological Technology (Hangzhou, China).

## Peptide synthesis

The following peptides (sCvsNPF: SQRSPSLRLRFamide; sPxsNPF-1: SVRSPSRRLRFamide; sPxsNPF-2: DTRQPVRLRFamide; sPxsNPF-3: SVRAPSMRLRFamide) were chemically synthesised by Sangon Biotech with a purity of >90%. The peptide was dissolved in double-distilled water at 5 mg/ml and stored at −80 °C. A fresh working solution needs to be prepared by diluting the storage solution with PBS.

## RNA sequencing and data analysis

The fat body and midgut of the 4 M instar *P. xylostella* larvae were dissected under the microscope in PBS on ice. Total RNA was isolated using an RNA isolater Total RNA Extraction Reagent (Vazyme) according to the manufacturer's instructions. The quality and the concentration of the isolated total RNA were estimated by electrophoresis and a NanoDrop 2000. The transcriptome sequencing was performed at Annoroad Gene Technology. cDNA libraries for transcriptome sequencing made from the total RNA were prepared using the NEBNext Ultra RNA Library Prep Kit for Illumina in conjunction with the NEBNext Poly (A) mRNA Magnetic Isolation Module. Libraries were validated and quantified before being pooled and sequenced on an Illumina HiSeq 2000 (Illumina) sequencer with a 150 bp paired-end protocol. The raw reads were filtered by removing adaptor sequences, empty reads and low-quality sequences (reads with unknown sequences 'N'). Sequences were de novo assembled using Trinity on a Galaxy Portal, and both ends were sequenced. The fragments per kilobase of transcript per million mapped reads (FPKM) method was used to analyse the expression profiles of all unigenes in different samples. Read counts were input into DESeq2 to calculate differential gene expression and statistical significance. Differentially expressed genes (DEGs) were screened using two criteria: (1) $\text{Log}_2$ (fold change) >1 and (2) a corresponding adjusted $P$ value less than 0.05. OmicShare (https://www.omicshare.com/tools/home/report/koenrich.html) was used to analyse the KEGG enrichment pathways.

## Quantitative PCR

To analyse gene expression in different tissues, *P. xylostella* larvae were dissected under the microscope, followed by the collection of the silk gland, fat body, midgut, haemocytes and central nervous system (CNS, brain and ventral nerve cord). Haemocytes were collected on clean Parafilm membranes with capillaries by bleeding the larvae from a cut proleg. To explore gene expression at different instars of parasitised or non-parasitised larvae, non-parasitised larva was isolated individually using RNA-easy Isolation Reagent (Vazyme). Parasitoid larvae were removed before parasitised hosts were isolated. As for the measurement of other experiments, per larva with different treatments was isolated individually using RNA-easy Isolation Reagent according to the manufacturer's instructions.

The quality and concentrations of total RNAs were estimated by electrophoresis and NanoDrop 2000 spectrophotometer (Thermo Fisher Scientific). First-strand cDNAs were synthesised using the ReverTra Ace qPCR RT kit (Toyobo) according to the manufacturer's instructions. Both β-actin (GenBank Acc. No.: AB282645) and β-tubulin (GenBank Acc. No.: EU127912) of *P. xylostella* were used as internal controls. Primer sequences used for qPCR analysis are shown in Appendix Table S4. qPCR was performed by using THUNDERBIRD® SYBR qPCR Mix (Toyobo) on the CFX Connect real-time system (Bio-Rad). All qPCR assays were performed using four internal replicates and at least three biological replicates under the following conditions: 95 °C for 60 s and 40 cycles of 95 °C for 15 s and 60 °C for 30 s. Relative gene expression levels were calculated using the $2^{-\Delta\Delta Ct}$ method.

## Collection of venom and CvBV particles

About 2-day-old female adults were used for the collection of CvBV and venom, as previously described (Wang et al, 2018; Yu et al, 2007). Wasp individuals were swabbed with 95% ethanol (v/v), and then the venom reservoirs were carefully removed from the abdomen of female wasps and placed into 40 µl prechilled PBS in a 0.5 ml Eppendorf tube on ice. The reservoirs were then torn open with forceps to release the venom and then centrifuged at 12,000×$g$ for 10 min at 4 °C. The venom supernatant was transferred to a clean Eppendorf tube and stored frozen at −80 °C. One venom reservoir equivalent (VRE) was defined as the supernatant from one torn venom reservoir in 1 µl PBS. To obtain CvBV particles, ovaries were dissected into prechilled PBS, and the calyx was punctured individually. The calyx fluid with PBS was filtered using a 0.22 µm filter to remove cellular debris, and centrifuged at 20,000×$g$ for 1 h. The viral particle pellet was resuspended in PBS and stored frozen at −80 °C. The viral particles collected from one single adult female were defined as one female equivalent (FE).

## Teratocyte collection

Teratocytes from *C. vestalis* were collected using previously established methods (Wang et al, 2018). Briefly, parasitised *P. xylostella* larvae were dissected in culture dishes containing TNM-FH medium (HyClone) plus antibiotics (ampicillin and kanamycin, each at 100 µg/l). The larvae were dissected to release the *C. vestalis* teratocytes into the medium. Then, 5-day-old teratocytes were collected and transferred to another dish containing medium.

Teratocytes were further washed five times in the medium, transferred to a microfuge tube and then gently centrifuged at 500×$g$ for 5 min. The resulting pellet consisted of only teratocytes whose abundance was estimated using a haemocytometer. The teratocytes collected from one single parasitised larva were defined as one teratocyte equivalent (TE). To obtain the teratocytes' contents, ~100 × 200 teratocytes were collected from parasitised *P. xylostella* larvae and transferred to 10 µl PBS. Teratocytes were broken using ceramic beads and centrifuged at 16,100×$g$ for 10 min at 4 °C to remove the debris. The supernatant was stored at −80 °C.

## Microinjection

Microinjections were performed under a Stemi 2000C microscope (Zeiss) using glass capillary injection needles and an Eppendorf Femtojet (Eppendorf) with a microcontroller (Narishige). To confirm the function of bovine insulin on hosts, 1 µg bovine insulin (Sigma-Aldrich, I0305000) was injected into the middle 3rd instar *P. xylostella* larvae. To identify the effect of each parasitic factor on hosts, 0.05 FE CvBV or 0.05 VRE venom was injected into the abdomen of the middle 3rd instar *P. xylostella* larva (Wang et al, 2018; Yu et al, 2007). 1 TE teratocytes' content was microinjected into the early 4th instar *P. xylostella* larvae. To confirm the function of each synthetic sNPF peptide, 0.005 ng, 0.05 ng, or 0.5 ng of each peptide was injected into the middle 3rd instar *P. xylostella* larvae in a volume of 0.1 µl/per larva once per day for 3 consecutive days.

## In vitro sNPF peptide and inhibitor treatment

The in vitro incubations of fat body or midgut were performed as previously described (Cheng et al, 2022). The fat bodies and midguts from non-parasitised middle 4th instar *P. xylostella* larvae were collected in 0.5 mL of PBS. The collected

tissues were then preincubated in culture dishes containing TNM-FH medium (HyClone) for 30 min. sNPF peptides or inhibitors were administered to isolated fat body and midgut in vitro using a modified method from a previous study (Oh et al, 2019). 1 mM stock solution of pertussis toxin (PTX, Tocris Bioscience) or U73122 (MedChemExpress) was dissolved in DMSO (Sigma-Aldrich) and kept at −80 °C. 0.1 ng/µl PTX or 10 µM U73122 were then mixed with 20 µM sNPF-contained TNM-FH medium. Preincubated tissues were incubated in these mixtures at 25 °C for 1 h, followed by total RNA isolation and qPCR.

## RNA interference

DNA fragments of ∼500 bp in size were amplified by PCR. Forward and reverse primers containing T7 promoter sequences at their 5'end were listed in Appendix Table S4. T7 High Yield RNA Transcription Kit (Vazyme) was used for the production and purification of double-stranded RNA (dsRNA) according to the manufacturer's instructions. A 328 bp coding sequence from green fluorescent protein (GFP, GenBank Acc. No.: AAB02574.1) was used as a control dsRNA (dsGFP). Synthetic dsRNA was confirmed by an agarose gel, and the concentration was determined by NanoDrop2000 spectrophotometer (Thermo Scientific). About 500 ng of dsRNA was injected into *P. xylostella* larvae 72 h post-parasitisation, and the RNAi efficiency was detected 48 h post-injection by qPCR. At least three biological replicates were performed.

## Western blotting and LC-MS/MS Analysis

For immunoblotting of CvsNPF, cell-free plasma of haemolymph from non-parasitised or parasitised 4L instar *P. xylostella* was collected after 4-fold dilution with cold PBS and centrifugation at 10,000×g for 10 min. For immunoblotting of AKT and phospho-AKT, the fat body and midgut of non-parasitised or parasitised 3 L instar *P. xylostella* were collected. For immunoblotting of PxsNPF and PxsNPFR, the midguts of non-parasitised 4 M instar *P. xylostella* were collected. Total protein from samples was extracted by Minute™ Total Protein Extraction Kit for Insects or Minute™ Plasma Membrane/Protein Isolation and Cell Fractionation Kit (Invent Biotechnologies) according to the manufacturer's protocol. Proteins were diluted in 5× Protein Sodium Dodecyl Sulfate Polyacrylamide Gel Electrophoresis Loading Buffer (Sangon Biotech), then boiled for 10 min. Proteins were separated by SDS-polyacrylamide gel electrophoresis (PAGE) and then transferred to a polyvinylidene difluoride (PVDF) membrane (Bio-Rad) using the Mini-ProTEAN Tetra system (Bio-Rad) at 12 V for 10 min.

PVDF membrane was incubated in a blocking solution (Tris-buffered saline containing 0.1% Tween 20, 5% BSA) for 1 h. Rabbit polyclonal anti-CvsNPF prepropeptide (generated against the peptides AENYLDYGEENADRNI by ABclonal; 1:500 dilution), rabbit polyclonal anti-PxsNPF prepropeptide (generated against the peptides QALSQYDSVAQSAQEAA, ABclonal, 1:500 dilution), rabbit polyclonal anti-PxsNPFR generated against the recombinant protein PxsNPFR (amino acids 1-42; ABclonal; 1:500 dilution), rabbit anti-phospho-AKT (Abmart, Cat# T40067, Ser473; 1:500 dilution) or rabbit anti-AKT (Huabio, Cat# ET1609-51, 1:500 dilution), mouse anti-β-Actin (ABclonal, 1:2000 dilution) and mouse anti-β-Tubulin (Fudebio, 1:1000 dilution) were

used as primary antibodies. The HRP Goat Anti-Rabbit IgG (ABclonal, 1:2000 dilution) or HRP Goat Anti-Mouse IgG (ABclonal, 1:2000 dilution) were used as the secondary antibody. After five washes with PBST, membranes were incubated with an ECL Chemiluminescence Kit (Vazyme) and imaged by Chemi Doc-It Imaging System (Bio-Rad).

To validate the specificity of polyclonal antibodies, the target band from SDS-PAGE was excised and subject to LC-MS/MS analysis for protein identification after proteolytic digestion by trypsin. Briefly, the treated sample was then subjected to a home-made C18 Nano-Trap column (Thermo Fisher, 164535, 5 cm × 75 µm, 3 µm). The separated peptides were analysed by Orbitrap Eclipse matched with FAIMS (Thermo Fisher). All resulting spectra were searched against target protein sequence by the search engines: Proteome Discoverer 3.1 (PD 3.1, Thermo). A maximum of 2 missed cleavage sites were allowed. In order to improve the quality of analysis results, the software PD 3.1 further filtered the retrieval results: Peptide Spectrum Matches (PSMs) with a credibility of more than 99% were identified PSMs. The identified PSMs and proteins were retained and performed with FDR no more than 1.0%. To validate the antibody specificity of anti-PxsNPF and anti-PxsNPFR, a band around 60 kDa was excised from the SDS-PAGE gel based on the western blot results (Appendix Fig. S10) and subjected to LC-MS/MS analysis. The detection of PxsNPF and PxsNPFR fragments were shown in Appendix Table S3, comfirmed the specificity of anti-PxsNPF and anti-PxsNPFR.

## Measurement of haemolymph carbohydrate levels by HPLC

For the determination of carbohydrate levels in the haemolymph of non-parasitised or parasitised *P. xylostella* larvae, high-performance liquid chromatography (HPLC) analysis was used as previously described with minor modifications (Mayack et al, 2020). 1 µL of haemolymph was collected, diluted 1000 times with double-distilled water and then heated at 95 °C for 5 min. Samples filtered through a 0.22-µm filter were used for HPLC analysis.

In all, 10 µl of supernatant was injected per sample. Carbohydrates were separated using a Dionex ICS-3000 Ion Chromatograph equipped with a Carbo PA1 Carbohydrate Column (Dionex) at 30 °C with a flow rate set at 1 ml/min. The mobile phase was NaOH at a concentration of 200 mM. Sample concentrations were calculated from the standard curve obtained from a serial dilution of the standard solution. Different sugar standards (Sigma-Aldrich), including sucrose, glucose, fructose, trehalose, sorbitol, arabinose, maltose, mannose, xylose and lactose, were mixed for HPLC analysis to identify each carbohydrate peak according to the typical retention times (Appendix Fig. S1A). Standard curves for each run were generated from the peak areas of the five standards (sucrose, glucose, fructose, trehalose, and sorbitol) with known sugar concentrations.

## Sugar measurement

For *P. xylostella* larvae, 1 µl of haemolymph from the moth was diluted 60-fold in cold PBS and subjected to trehalose determination. The fat body of *P. xylostella* larvae was homogenised in 200 µl of cold PBS. Half of the homogenate was centrifuged at 13,000 rpm for 15 min, and the supernatant was used for protein content

determination by Bradford Reagent (Invitrogen). The rest was heated at 70 °C for 5 min followed by centrifuging at 13,000 rpm for 15 min, and the supernatant was used for glycogen determination.

For *C. vestalis* larvae, ~20 wasp larvae were collected from parasitised 4 L instar *P. xylostella* and homogenised with 400 μL PBS. Half of the homogenate was centrifuged at 13,000 rpm for 15 min, and the supernatant was quantified for protein content using the Bradford assay described above. The rest was heated at 70 °C for 5 min followed by centrifuging at 13,000 rpm for 15 min, and the supernatant was used for trehalose, glycogen and glucose determination, respectively.

Trehalose content determination was performed by the anthrone sulfate method. 15 μL of the sample was incubated with 15 μL of 1% $H_2SO_4$ at 90 °C for 10 min, then 15 μL of 30% KOH was added and incubated at 90 °C for 10 min. Finally, 300 μl of 0.2% anthrone (Sigma-Aldrich) solution in 96% $H_2SO_4$ was added and incubated at 90 °C for 10 min. All reactions were stopped by immersion in ice. To create a trehalose standard curve, 15 μl of trehalose standards (Sigma-Aldrich) were treated similarly as a sample. Finally, 100 μl of each sample was added to a 96-well plate. The absorbance of samples was then measured at 630 nm on a SpectraMax® iD5 Multi-Mode Microplate Reader (Molecular Devices), and the trehalose levels were calculated based on the standard curve and normalised by protein content or sample volume.

For glycogen measurement, 50 μL of heat-treated homogenate was incubated with either 50 μL of amyloglucosidase (Sigma-Aldrich) or 50 μl of PBS. To create a glycogen standard curve, 50 μL of glycogen standards (Sangon Biotech) were treated with either 50 μL of amyloglucosidase or 50 μL of PBS. All samples were incubated at 37 °C for 1 h. Then, 30 μL of each sample was added to a 96-well plate. Next, samples were determined by using the GAHK20 Glucose Assay kit (Sigma-Aldrich), according to the manufacturer's directions and normalised by protein content. The absorbance of samples was then measured at 340 nm and normalised by subtracting the absorbance of the free glucose of untreated samples from the absorbance of the total amount of glucose present in samples treated with amyloglucosidase. Glycogen content was then calculated based on the glycogen standard curve. Glucose levels were determined by the GAHK20 Glucose Assay kit (GAHK20) according to the manufacturer's directions and normalised by protein content.

## Measurement of trehalase activity

Trehalase activity was detected as previously described with minor modifications (Shi et al, 2016). *P. xylostella* larvae at different instars were homogenised in 200 μL of cold PBS. The homogenate was then centrifuged at 12,000×*g* for 10 min at 4 °C. The supernatant was directly used to measure the soluble trehalase activity. The reaction mixture (250 μL) consisted of 62.5 μL of 0.04 M trehalose (Sigma-Aldrich), 50 μL of the supernatant, and 137.5 μL of PBS. The mixture was incubated at 37 °C for 30 min, and the reaction was stopped by heating at 95 °C for 5 min. The trehalase activity was based on the rate of glucose released from trehalose, measured using a Glucose (GO) Assay Kit (Sigma-Aldrich). The protein amount in each sample was estimated as described above. Trehalase activity was expressed as μg (glucose) $mg^{-1}$ (protein) $min^{-1}$.

## Glucose 6-phosphate determination

Glucose 6-phosphate (G6P) was determined using a G6P kit (Sigma-Aldrich) following the manufacturer's instructions. The fat body of *P. xylostella* larvae was homogenised in 200 μL of cold PBS. The homogenate was centrifuged at 13,000×*g* for 15 min to remove insoluble material, and then the supernatant was collected. In all, 15 μL of the supernatant was quantified for protein content determination using the Bradford assay described above. The remaining supernatant was deproteinized with a 10 kDa MWCO spin filter (Merck). Then, 30 μL of sample and 20 μL of G6P assay buffer were added to duplicate wells of a 96-well plate, and 50 ul of reaction mixture containing 46 μL of G6P assay buffer, 2 μL of G6P enzyme, and 2 μL of G6P substrate was applied to each well and carefully mixed. Triplicate assays were set up for each sample. In addition, a blank sample was set up by omitting the G6P enzyme mix to remove the effect of NADH or NADPH background. The blank readings were then subtracted from the sample readings. Finally, the plate was protected from light and incubated at room temperature for 30 min. Absorbance was measured at 450 nm. The amount of G6P (nmole) was calculated based on the standard curve. The concentration of G6P was determined using the following equation and normalised by protein content: amount of G6P/sample volume.

## Glycogen staining

Staining glycogen was performed using the glycogen periodic acid schiff (PAS/Hematoxylin) stain kit (Solarbio) according to the manufacturer's instructions. In brief, the fat bodies of *P. xylostella* larvae were collected and fixed with 4% paraformaldehyde for 30 min, washed twice with PBS, incubated with buffer A (periodic acid solution) for 5 min, and washed twice with PBS. Samples were stained with buffer B (Schiff Reagent) for 10 min, and washed with PBS for 10 min. For staining nuclei, buffer C (Hematoxylin solution) was applied for 1 min and transferred with buffer D (Eosin solution). Then samples were washed with PBS for 15 min and mounted in gold antifade mountant (Invitrogen). Images were acquired with a Zeiss Primo Star microscope equipped with a Zeiss AxioCam ERc camera, except for ds*AKHR*- and dsGFP-treated samples (KEYENCE VHX-2000C).

## Cyclic AMP and intracellular Ca²⁺ Assay

To establish a stable HEK293/PxsNPFR cell line, the pcDNA 3.1$^{(+)}$/*PxsNPFR* plasmid was transfected into the HEK293 cells using Lipofectamine 3000 (Invitrogen). Two days later, 800 μg/ml G418 (Sangon Biotech) was added to the culture medium to select cells that stably expressed the receptor. After three weeks, resistant clones were trypsinised in cloning cylinders and transferred to 12-well plastic plates for expansion.

Measurement of cyclic AMP in the HEK293/PxsNPFR cell line was accomplished with the cAMP-Glo™ Assay (Promega) according to the manufacturer's instructions. Cells were seeded into the 96-well tissue culture plate (Thermo) overnight to grow at a recommended cell density (2500-10,000 cells/well). For the assay, cells were washed three times with PBS and then stimulated with 10 μM forskolin (Beyotime, added with 500 μM IBMX, Sigma-Aldrich) alone or 10 μM forskolin (added with 500 μM IBMX) with

the indicated concentrations of synthetic sNPF for 15 min at room temperature. To generate a cAMP standard curve, cAMP standards supplied with the kit were prepared in a separate 96-well. Chemiluminescent signals were measured for 1 s/well on a SpectraMax iD5 Multi-Mode Microplate Reader (Molecular Devices).

To measure the intracellular calcium flux, the transfected cells HEK293/PxsNPFR were harvested with PBS solution containing 0.02% EDTA, rinsed twice with PBS, and resuspended in Hanks' balanced salt solution containing 0.025% BSA. Cells were then loaded with 2.5 µM Fura-2/AM (Dojindo) for 30 min at 37 °C and washed twice in Hanks' balanced saline solution. These cells were stimulated with the indicated concentrations of synthetic sNPF, and Hanks' solution was used as a negative control. The calcium flux was measured using excitation at 340 nm and 380 nm on a SpectraMax iD5 Multi-Mode Microplate Reader (Molecular Devices) at 37 °C.

## Immunohistochemistry

To localise PxsNPF prepropeptide and PxsNPFR, the midgut and brain of non-parasitised or parasitised *P. xylostella* at 3L instar and 4M instar were collected as described above. To localise CvsNPF prepropeptide, 3-day-old teratocytes and the brain of late 2nd instar *C. vestalis* larvae were collected. To localise PxsNPFR in the HEK293/PxsNPFR cell line, the cells were washed with PBS carefully and then plated into a chamber slide (Thermo).

Samples were fixed in a 4% paraformaldehyde solution for 30 min. After rinsing in PBST, tissues were blocked with 5% BSA in PBST for 2 h and incubated overnight at 4 °C with the primary antibody. The following primary antibodies were used: rabbit polyclonal anti-PxsNPF prepropeptide (generated against the peptides QALSQYDSVAQSAQEAA, ABclonal, 1:200 dilution), rabbit polyclonal anti-CvsNPF prepropeptide (generated against the peptides AENYLDYGEENADRNI, ABclonal, 1:200 dilution) or rabbit polyclonal anti-PxsNPFR generated against the recombinant protein PxsNPFR (amino acids 1-42; ABclonal; 1:200 dilution). Samples were then washed three times in PBST and incubated with the following secondary antibody for 2 h. Alexa Fluor 488 (Invitrogen, 1:1000 dilution) or Alexa Fluor 594 (Invitrogen, 1:1000 dilution) was used as the secondary antibody. CellMask™ Plasma Membrane Stains (Invitrogen) were used for the membrane staining by incubation for 30 min. The nuclei were stained with DAPI (Roche) for 5 min. The specimens were mounted in Gold Antifade Mountant (Invitrogen). Images were visualised using a Zeiss LSM 800 confocal microscope and analysed with ZEN imaging software v.2.3 Lite (ZEISS) or ImageJ 2.9.0 (National Institutes of Health). For fluorescence intensity quantification, corrected total cell fluorescence was calculated by subtracting the mean background fluorescence from the mean fluorescence intensity of each selected region.

## Food intake assay

Quantification of food ingestion was performed as previously described (Zhan et al, 2016). Briefly, after fasting for 30 min, every 10 *P. xylostella* larvae were allowed to transfer into a dish and fed on an artificial diet that contained 0.5% (w/v) blue food dye (Sigma-Aldrich). Feeding was interrupted at different time points

by freezing the dishes at −80 °C. Larvae were homogenised in 300 µl PBS buffer with 1% Triton X-100 and centrifuged at 13,000 rpm for 30 min to clear the debris. The absorbance of the supernatant was measured at 625 nm. The background absorbance for supernatants was from larvae fed with regular food. The net absorbance reflected the amount of food ingested.

## Biological characterisation of parasitoid offspring

Wasp larvae from the 4 M instar parasitised *P. xylostella* larvae with dsRNA injection were collected. A KEYENCE VHX-2000C microscope was used to photograph wasp larvae and measure the head capsule width. The head capsule (viewed dorsally) of per larva was measured transversely across the eyes at the broadest point of head width.

The pupation and eclosion rate were calculated as the percentage of wasps that successfully pupated and eclosed from the total of parasitised hosts. The pupation and eclosion time of wasps emerging from parasitised *P. xylostella* larvae were recorded at 24 h intervals.

Wasps from parasitised *P. xylostella* larvae injected with dsRNA were divided into two groups after eclosion. In the mated group, females and males were kept together for 48 h to mate sufficiently; in the virgin group, female and male wasps were kept separately to prevent the females from mating. Each box contained 40 3M instar *P. xylostella* and one mated or virgin female wasp. The females were removed after 1 h of parasitism. Wasp eggs of the hosts were collected 24 h after parasitism. The oviposition ability of each female wasp was assessed by the number of eggs laid in 1 h.

The wasp cocoon from the host injected with dsRNA was collected and reared separately. They were then divided into two groups according to diet. In the sugar-diet group, wasps were fed with 20% (v/v) honey solution. In the water-diet group, wasps were fed with water-only. The survival days of adult wasps from eclosion to death were recorded every 24 h.

## Quantification and statistical analysis

ImageJ 2.9.0 was used for quantitative comparison of positive signals in immunofluorescent staining images. Data are presented as mean ± standard error of the mean (SEM). Statistical analyses were performed with GraphPad Prism 9.0. Comparisons between two groups were performed using two-tailed unpaired Student's $t$ tests; comparisons across multiple groups were assessed using one-way ANOVA and two-way ANOVA (used only if there were more than one variant). The post hoc test with Tukey's correction was performed for multiple comparisons following ANOVA. The statistical methods of sequencing data were described in the figure legends. The sample numbers ($n$) are shown in the figure legends. $P < 0.05$ indicated the differences were statistically significant. The exact $P$ values of each figure are provided in Appendix Table S5.

# Data availability

The raw transcriptomic data for the fat body, midgut and CNS of *P. xylostella* were deposited to BioProject under the accession number PRJNA1073706, and the lists of DEG were included in Appendix Datasets EV1 and 2. All other data are included in the main text and/or supporting information.

The source data of this paper are collected in the following database record: biostudies:S-SCDT-10_1038-S44318-025-00636-5.

## Peer review information

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

## Acknowledgements

The authors would like to thank all study participants, as well as Xiqian Ye (the Instrument Service Platform of Zhejiang University Institute of Insect Sciences) and Youping Xu (the Analysis Center of Agrobiology and Environmental Sciences, Zhejiang University) for their technical support. We also thank Nai-Ming Zhou (Zhejiang University) for providing pertussis toxin. This research was funded by the National Natural Science Foundation of China (U22A20485 and 32272607), the National Key R&D Program of China (2023YFD1400800) and the Fundamental Research Funds for the Central Universities (226-2024-00070).

## Author contributions

**Zhi-Zhi Wang**: Supervision; Funding acquisition; Methodology; Writing—original draft; Writing—review and editing. **Ruo-Fei Ma**: Resources; Investigation; Methodology; Writing—original draft. **Li-Cheng Gu**: Resources; Investigation. **Li-Zhi Wang**: Resources; Investigation. **Ting Chen**: Investigation. **Pei Yang**: Investigation. **Jia-Ni Zou**: Investigation. **Jiang-Yan Zhu**: Investigation. **Zhi-Wei Wu**: Investigation. **Yue-Nan Zhou**: Resources. **Min Shi**: Methodology; Writing—original draft. **Xing-Xing Shen**: Writing—review and editing. **Jian -Hua Huang**: Supervision; Writing—review and editing. **Xue-Xin Chen**: Conceptualisation; Supervision; Funding acquisition; Writing—review and editing.

Source data underlying figure panels in this paper may have individual authorship assigned. Where available, figure panel/source data authorship is listed in the following database record: biostudies:S-SCDT-10_1038-S44318-025-00636-5.

## Disclosure and competing interests statement

The authors declare no competing interests.

