## [Peer Review File · The EMBO Journal]

Dual interference with host neuropeptide signaling allows parasitoid wasp to hijack host sugar metabolism

Zhi-Zhi Wang, Ruo-Fei Ma, Li-Cheng Gu, Li-Zhi Wang, Ting Chen, Pei Yang, Jia-Ni Zou, Jiang-Yan Zhu, Zhi-Wei Wu, Yue-Nan Zhou, Min Shi, Xing-Xing Shen, Jianhua Huang, and Xue-Xin Chen

Corresponding author: Xue-Xin Chen (xxchen@zju.edu.cn)

Review Timeline:

Submission Date:	25th Feb 25
Editorial Decision:	7th Apr 25
Revision Received:	30th Aug 25
Editorial Decision:	26th Sep 25
Revision Received:	16th Oct 25
Accepted:	27th Oct 25

Editor: Ieva Gailite

Transaction Report:

Dear Dr. Chen,

Thank you for submitting your manuscript for consideration by the EMBO Journal. We have now received comments from a full set of reviewers, which are included below for your information.

As you will see, all reviewers are generally positive in their assessment and appreciate the contribution of the study to the research field. At the same time, they indicate a number of substantive concerns that would be important to address in the revised study, in particular asking for further analysis of potential contribution of insulin signalling to the observed parasite-induced increase in trehalose levels. They further find that further insights into parasite- vs host-derived sNPF signalling will be needed. Finally, reviewer #3 requests further insights into the physiological impact of the reported changes in host metabolism and sNPF signalling for parasitoid wasp development and fecundity.

Based on the interest expressed in the reports, I would like to invite you to address these comments in a revised manuscript. I think that it would be useful to discuss the revision in more detail via email or phone/videoconferencing - please let me know which option you prefer.

We generally allow three months as standard revision time, which can be extended to six months in the case of major revisions. Should you foresee a problem in meeting this deadline, please let us know in advance to discuss an extension. As a matter of policy, competing manuscripts published during this period will not negatively impact on our assessment of the conceptual advance presented by your study. However, please contact me as soon as possible upon publication of any related work to discuss the appropriate course of action.

When preparing your letter of response to the referees' comments, please bear in mind that this will form part of the Review Process File and will therefore be available online to the community. For more details on our Transparent Editorial Process, please visit our website: <https://www.embopress.org/page/journal/14602075/authorguide#transparentprocess>. Please also see the attached instructions for further guidelines on preparation of the revised manuscript.

Please feel free to contact me if have any further questions regarding the revision. Thank you for the opportunity to consider your work for publication, and I look forward to discussing your revision with you.

With best wishes,

Ieva Gailite

- a point-by-point response to the referees' comments, with a detailed description of the changes made (as a word file).

- a word file of the manuscript text.
 - individual production quality figure files (one file per figure)
 - a complete author checklist, which you can download from our author guidelines (<https://www.embopress.org/page/journal/14602075/authorguide>).
 - Expanded View files (replacing Supplementary Information)
- Please see out instructions to authors
<https://www.embopress.org/page/journal/14602075/authorguide#expandedview>
- a Reagents and Tools Table as part of the Methods section, which can be downloaded from our author guidelines (<https://www.embopress.org/page/journal/14602075/authorguide#structuredmethods>)

We realize that it is difficult to revise to a specific deadline. In the interest of protecting the conceptual advance provided by the work, we recommend a revision within 3 months (6th Jul 2025). Please discuss the revision progress ahead of this time with the editor if you require more time to complete the revisions.

Referee #1:

The authors demonstrate that parasitism affects the host trehalose level by increasing glycogenolysis and glycolysis. Then, they showed that host hypertrehalosemia is essential for parasite development. This parasite-induced hypertrehalosemia is not affected by insulin and AKH signaling, as well as food intake, but affected by the evolutionary conserved short neuropeptide F (sNPF). Parasite teratocytes produced sNPF binds host sNPFR and activates Gi/cAMP pathway, which promotes glycogenolysis in the fat body. On the contrary, parasite virus-induced host midgut trachea sNPF binds host sNPFR and activates Gq/Ca2+ pathway, which activates glycolysis in the midgut. In the structure analysis, the host sNPFR has different binding pockets to the parasite sNPF and the host sNPF.

This manuscript shows valuable insights into the parasite-host carbohydrate metabolism.

Major comments:

1. The authors insist that Insulin-PI3K/AKT signaling is independent of the parasitism-induced hypertrehalosemia. However, in Figure 3B, ILP3 mRNA level was changed in the CNS and the fat body after parasitism. In Figure 3C, the immunoblots need quantitation. AKT/pAKT levels of non-parasitism and parasitism in the fat body and the midgut do not look the same. Therefore, Insulin-PI3K/AKT signaling can't be excluded.
2. How does a parasite virus induce host sNPF in the midgut trachea?
3. sNPFR signaling in the CNS is a major source for regulating food intake, which modulates the carbohydrate metabolism systemically. The authors did not mention any role of sNPFR signaling in the CNS by the parasite sNPF and host sNPF.
4. One can expect parasite sNPF and host sNPF expression simultaneously in parasitism. What is the food intake phenotype of sPxsNPF1+sCvsNPF peptides co-injection? Why is sPxsNPF1 peptide, not sCvsNPF peptide, injection reduced food intake like parasitism?

Minor comments:

1. In lane 83, Fig.1D to FigS1D.
2. In lane 1013, the first antibody to the primary antibody.

Referee #2:

Carbohydrate metabolism has been previously identified as an important determinant of host-parasite infections, and this manuscript by Wang et al., seeks to understand the basis of this observation using the *Cotesia vestalis* - *Plutella xylostella* system. They find that the parasitoid manipulates host signaling via two discrete mechanisms: an infection induced manipulation of host hormone signaling, and a teratocyte derived parasitoid hormone that cross-reacts with the host receptor.

The authors further use pharmacological and reverse genetics approaches to map out the basis of these responses: the manipulation of expression of transcripts encoding specific carbohydrate metabolic enzymes.

Overall, this is a very interesting and thorough study of an important topic in host-parasite infection biology. With one exception (below), I feel that the conclusions of the study are robustly supported by the data, and by and the large, the manuscript is easily understood. I also want to commend the authors on including a clear and reproducible methods section (with one exception, noted below).

There is only one aspect of the paper that I felt was not supported by adequate experimentation and analysis. The authors state that the protein levels of AKT and phospho-AKT were unchanged following infection (lines 138-140, and Fig 3C). The only data shown to support this are from a single replicate Western blot. There doesn't appear to be any replication or quantification performed, and I don't see any evidence that the loading control was considered. The image itself is also not within the linear range of the signal (for instance, the fat body loading control is saturated, and so not quantifiable). The overall findings for a lack of insulin signaling involvement in the phenomenon are supported by other data, but this is a weak spot within the argument.

Overall the manuscript is well written and easy to follow, but I do have several minor concerns that should be addressed for clarity:

1. The justification for using bovine insulin needs to be strengthened. It would be good to have a short summary of the existing data, including for instance, which species this experiment has been conducted in and the data linking bovine insulin to activation of insect insulin-like pathways. As is, it is a bit difficult to interpret these results.
2. The results section concerning the CvsNPF antibody doesn't quite make sense (lines 178-180). The experiment was set up as a test of whether CvsNPF is secreted into the host, and presumably interacts with host tissues. These data would be a major strength of the study, however, the only data presented are that teratocytes express the peptide.
3. The molecular docking section (lines 286-308) should be more specific that the structures and interactions discussed are predictions rather than experimental data. In interpreted the manuscript as reporting experimentally derived structures of the NPF and NFR proteins, with simulated docking. However, from the methods, I think that the structures are AlphaFold predictions instead.
4. I have some concerns about the immunofluorescence staining intensity analysis. The methods don't mention any form of normalization/control for fluorescence background. This is standard for fluorescence intensity analysis (i.e. the corrected total cell fluorescence [CTCF] approach), unless of course the data collected were photon counts, or similar. The methods section should be updated to address this issue, and the data reanalyzed if necessary (which should be fairly straightforward using typical approaches).

I noticed several typos throughout the manuscript, which I'm sure will be caught in proofreading, so no real issue at this stage but please make sure to double check everything, in particular references to figure panels.

Referee #3:

In Wang et al., the authors focus on the relationship between the metabolism of the host *Plutella xylostella* and the parasitoid wasp *Cotesia vestalis* and report that parasitoid wasps regulate host metabolism through manipulation of the neuropeptide sNPF. Although some regulation of host behavior by parasitoids has been known, there are few examples of parasitoids' metabolic control through hijacking neuropeptide signaling, which readers will find interesting. This paper's message is derived from various experiments, including gene knockdown, peptide injections, and molecular simulations. However, some links between host trehalose metabolism and parasite development are missing. Moreover, the effects of neuropeptides on the host also need to be scrutinized in terms of parasite development and methods. Therefore, I argue for additional experiments (Major concern) and revisions (Minor concern).

Major concern

Host sugar metabolism and parasitoid wasp growth

1. It is unclear how the lack of an increase in blood trehalose in the host *Plutella xylostella* would affect the metabolism of the parasitoid wasp *Cotesia vestalis*. Since the decrease in trehalose due to pxGP knockdown is slight (roughly 20 $\mu\text{g}/\mu\text{L}$ hemolymph, it is even higher than non-parasite), it is unlikely that this slight decrease would affect the development of parasitoid wasps. Rather, parasitism or knockdown of pxGP may cause changes in the amounts of G6P or other intermediate metabolites of the glycolytic system, metabolites of the TCA circuit, and amino acids, which may affect the development of parasitoid wasps. To examine that trehalose is, in fact, the molecule that supports parasitoid bee growth, knockdown of TPP and TPS should be performed to investigate the relationship between trehalose synthesis and parasitoid wasp growth.

2. Concerning Fig. 2F, we have found that fecundity is reduced in female parasitoid wasps, but is this restored by sugar feeding, like the reduced starvation tolerance seen in males? It is necessary to determine whether the reduction of glycogen after eclosion is directly related to the number of eggs laid.

Validation of sNPF and sNPFR signaling

3. There is no evidence in the current manuscript that hijacking sNPF/sNPFR signaling aids parasitic wasp development. Therefore, the success of parasitoid wasp parasitism against sNPFR knockdown hosts should be examined, as shown in Fig. 2.

4. Since no validation of the CvsNPF antibody has been done, more data is needed to support the evidence that CvsNPF is expressed in Teratocytes. The expression of sNPF in Teratocytes should be validated by qPCR or Western blot using Teratocytes. In addition, I could not find any data on sNPF expression in teratocytes from the manuscript cited in L171-175. If the expression of sNPF were detected in a previous study (Gao et al., 2016), it would be preferable to reanalyze and show the data of sNPF expression from the data of previous studies.

5. Regarding Fig. 5E, looking at the source data for the image, there appears to be an increase in what looks like round cells expressing sNPF. Are these cells tracheal cells, hemocytes, or enteroendocrine cells? These cells should be mentioned and discussed to better understand the phenomenon. Furthermore, since antibody staining sometimes detects many nonspecific signals, validation that the cells stained with antibodies are sNPF should be performed by the knockdown of sNPF.

6. The image in Fig. S6B suggests that sNPF acts autocrine on sNPFR in the trachea. Is hexokinase expressed in the intestinal trachea? It is worth discussing whether the trachea regulates whole-gut metabolism or an alternative possibility. As for the sNPFR antibodies, since antibodies against GPCRs tend to be challenging to produce, it would be desirable to validate whether the signals seen in Fig. S6B are due to sNPFR using Western blotting or sNPFR knockdown.

7. Since Px-sNPFs mature from a common prepro-sNPF via processing, the host cells should produce not only sNPF-1 but also sNPF-2 and 3 together. However, sNPF-2, 3 did not cause an increase in trehalose like sNPF-1 as in Fig.5. Can host sNPFR distinguish between sNPF-1, 2, and 3? Discussing the differences in the actions of sNPF-2, 3 as well as CvsNPF and sNPF-1 would enhance readers' understanding of neuropeptide actions in the future.

Minor concern

8. Regarding the heatmap of RNA-seq in Fig. 3B, showing the variation of gene expression of NP and P between different tissues highlights the differences between tissues rather than between NP and P. The heatmap should be separated for each tissue to show the variation in gene expression between NP and P.

9. The authors examine the expression of InR in Fig.3D, and it appears to be upregulated in P compared to NP. This is not seen in the RNA-seq heatmap in Fig. 3B. Elevated InR expression is a typical phenotype of insulin resistance (Pasco and Leopold, 2012, PMID: 22567167), suggesting that hyperglycemia is causing insulin resistance in host larvae. Indeed, bovine insulin fails to cause a decrease in trehalose at P (Fig. 3F), which is also a typical phenotype of insulin resistance. Since insulin resistance further exacerbates hyperglycemia, insulin resistance may be involved in the host hyperglycemic state found by the authors, and the current claim of insulin signaling-independent needs to be changed.

10. Regarding Fig. 3C, are all bands of pAkt and AKT detecting PxAKT? AKT is around 60 kDa in many organisms, and whether the band in Fig. 3C, which is 70 kDa, detects AKT is questionable. The predicted Mw of a protein (XP_048483457.1) that appears to be an AKT homolog of *Plutella xylostella* is around 60 kDa. In some cases, such as in *Drosophila*, there is a large isoform, and two bands are detected (see PMID: 20333234, Fig. 6), but it is necessary to check the blotting bands against the actual size of the protein.

11. Which tissue is used for qPCR data, such as Fig. 5A? If it is a whole body, it should be stated in the legend or the text. Also, it is better to describe the tissues used for the data and the instar of the larvae in the legend throughout the paper to help readers better understand the data.

12. The method does not describe the solvent and volume of injection (nL~ μ L) for microinjection of sNPF. This information should be included for researchers who will conduct similar experiments.

13. Legend: Fig. 3F and 3E are reversed.

14. L574-576: Immuno blot data for CvsNPF does not appear to be included in the manuscript.

15. L710: The sequence used to create the sNPFR antibody should be described

16. Source data: Fig.3D-3F is missing

Responses to Reviewers:**Referee #1:**

The authors demonstrate that parasitism affects the host trehalose level by increasing glycogenolysis and glycolysis. Then, they showed that host hypertrehalosemia is essential for parasite development. This parasite-induced hypertrehalosemia is not affected by insulin and AKH signaling, as well as food intake, but affected by the evolutionary conserved short neuropeptide F (sNPF). Parasite teratocytes produced sNPF binds host sNPFR and activates Gi/cAMP pathway, which promotes glycogenolysis in the fat body. On the contrary, parasite virus-induced host midgut trachea sNPF binds host sNPFR and activates Gq/Ca²⁺ pathway, which activates glycolysis in the midgut. In the structure analysis, the host sNPFR has different binding pockets to the parasite sNPF and the host sNPF.

This manuscript shows valuable insights into the parasite-host carbohydrate metabolism.

Response: We highly appreciate your insightful comment and your interest in the mechanistic role of parasitoid-derived viruses.

Major comments:

1. The authors insist that Insulin-PI3K/AKT signaling is independent of the parasitism-induced hypertrehalosemia. However, in Figure 3B, ILP3 mRNA level was changed in the CNS and the fat body after parasitism. In Figure 3C, the immunoblots need quantitation. AKT/pAKT levels of non-parasitism and parasitism in the fat body and the midgut do not look the same. Therefore, Insulin-PI3K/AKT signaling can't be excluded.

Response: To better analyze the Insulin-PI3K/AKT signalling pathway, the levels of AKT and pAKT were measured. The results showed that neither AKT nor pAKT levels changed in the fat body or the midgut after parasitism (**Figure R1**). Moreover, we agree that the Insulin-PI3K/AKT signalling pathway cannot be excluded. As noted by the other reviewers, parasitism-induced insulin resistance may partially contribute to hypertrehalosemia. Accordingly, we have revised some of our statements in the manuscript.

Figure R1. (A) Immunoblot analysis of total AKT, phosphorylated (pS473) AKT (p-AKT) and β -actin (loading control) in the fat body and midgut in nonparasitized and parasitized host larvae (late stage of 3rd instar). (B) The relative protein level of AKT/ β -actin (left) and p-AKT/AKT (right) was measured by Image J. The Bars represent mean \pm SEM of triplicates. Two-way ANOVA and Tukey's post hoc test (Not shown: $p > 0.05$).

2. How does a parasite virus induce host sNPF in the midgut trachea?

Response: We acknowledge that the potential regulatory mechanisms of parasitoid symbiotic virus in the host were not sufficiently explored in our initial manuscript. Although our study does not yet provide direct mechanistic evidence, previous findings demonstrated that insect tracheal epithelial

cells serve as a major conduit for systemic viral dissemination (PMID: 8159729). Building on this, we propose that a symbiotic virus may infect host tracheal cells and integrate intraparticle viral fragments into the genome of tracheal cells, which could directly modulate host gene expression, such as *sNPF*. Combining with the other reviewers' comments, we have now included a discussion of this in the revised version (**Lines 373-380**).

3. sNPFR signaling in the CNS is a major source for regulating food intake, which modulates the carbohydrate metabolism systemically. The authors did not mention any role of sNPFR signaling in the CNS by the parasite sNPF and host sNPF.

Response: We discussed the potential role of sNPFR signalling within the central nervous system in modulating carbohydrate metabolism in the previous version (**Lines 406-411**). Our results suggest that parasitism-induced hypertrehalosemia occurs independently of changes in food intake. Although the role of sNPFR signalling in promoting feeding behaviour is well established in *Drosophila* (PMID: 22876196), its function appears to differ in *P. xylostella*, where sNPF has been observed to suppress feeding. Whether the underlying molecular mechanisms are conserved between these species remains unclear. Thus, in the present study, we did not specifically address the potential involvement of central sNPFR signalling in the regulation of food intake.

4. One can expect parasite sNPF and host sNPF expression simultaneously in parasitism. What is the food intake phenotype of sPxsNPF1+sCvsNPF peptides co-injection? Why is sPxsNPF1 peptide, not sCvsNPF peptide, injection reduced food intake like parasitism?

Response: An equal volume mixture of sPxsNPF1 and sCvsNPF peptides was injected into host larvae. The food intake assay results indicated that co-injection of these peptides did not alter the feeding phenotype (**Figure R2**). We suppose that the effect of PxsNPF-1 on food intake is dose-dependent, similar to its effect on host trehalose levels (**Fig. 5B**).

Regarding the differential effects of sPxsNPF1 and sCvsNPF peptides on reducing food intake, the variation may be attributed to differences in the binding affinities between the ligands and their respective receptors. A similar phenomenon has been observed in *Bombyx mori*. Despite the co-expression of multiple sNPF isoforms, only sNPF-2 significantly enhanced feeding behaviour, while sNPF-1 and sNPF-3 were ineffective (PMID: 31116539). This functional specificity is largely supported by structural prediction and receptor docking analysis (**Fig. 6I**). Although both peptides share the conserved C-terminal RFamide motif, their overall 3D conformations and binding orientations within the receptor differ significantly.

Figure R2. Food consumption in middle 4th instar *P. xylostella* larvae injected with sPxsNPF-1, sCvsNPF, or sPxsNPF1+sCvsNPF peptides. Larvae were injected with a synthetic peptide or sPxsNPF1+ sCvsNPF mixture at 0.5

ng/larva. Comparisons between the sPxsNPF1-injected group and sPxsNPF1+sCvsNPF co-injected group were performed using two-way ANOVA and Tukey's post hoc test (* $p < 0.05$, ** $p < 0.01$, *** $p < 0.005$, Not shown, $p > 0.05$).

Minor comments:

1. In lane 83, Fig.1D to FigS1D.

Response: Corrected.

2. In lane 1013, the first antibody to the primary antibody.

Response: Corrected.

Referee #2:

Carbohydrate metabolism has been previously identified as an important determinant of host-parasite infections, and this manuscript by Wang et al., seeks to understand the basis of this observation using the *Cotesia vestalis* - *Plutella xylostella* system. They find that the parasitoid manipulates host signaling via two discrete mechanisms: an infection induced manipulation of host hormone signaling, and a teratocyte derived parasitoid hormone that cross-reacts with the host receptor.

The authors further use pharmacological and reverse genetics approaches to map out the basis of these responses: the manipulation of expression of transcripts encoding specific carbohydrate metabolic enzymes.

Overall, this is a very interesting and thorough study of an important topic in host-parasite infection biology. With one exception (below), I feel that the conclusions of the study are robustly supported by the data, and by and the large, the manuscript is easily understood. I also want to commend the authors on including a clear and reproducible methods section (with one exception, noted below).

There is only one aspect of the paper that I felt was not supported by adequate experimentation and analysis. The authors state that the protein levels of AKT and phospho-AKT were unchanged following infection (lines 138-140, and Fig 3C). The only data shown to support this are from a single replicate Western blot. There doesn't appear to be any replication or quantification performed, and I don't see any evidence that the loading control was considered. The image itself is also not within the linear range of the signal (for instance, the fat body loading control is saturated, and so not quantifiable). The overall findings for a lack of insulin signaling involvement in the phenomenon are supported by other data, but this is a weak spot within the argument.

Response: Thank you very much for the positive feedback and the valuable suggestions and comments. We have now included the quantitative analyses of the protein levels of AKT and phospho-AKT (with three biological replicates) in the revised manuscript. The detailed loading control information was included in the legend of **Fig. 3C**.

Overall the manuscript is well written and easy to follow, but I do have several minor concerns that should be addressed for clarity:

1. The justification for using bovine insulin needs to be strengthened. It would be good to have a

short summary of the existing data, including for instance, which species this experiment has been conducted in and the data linking bovine insulin to activation of insect insulin-like pathways. As is, it is a bit difficult to interpret these results.

Response: Thank you for pointing out the need to better justify the use of bovine insulin in our experimental design. We have revised the manuscript to include supporting evidence demonstrating that bovine insulin has been widely used in insect studies as a functional analogue of insect insulin-like peptides (ILPs), due to its ability to activate conserved insulin signalling pathways.

Evidence from multiple Lepidopteran insects demonstrates its effectiveness in mimicking endogenous insulin-like peptides (ILPs). For instance, in *Spodoptera exigua* and *Antheraea pernyi*, bovine insulin injection led to a significant reduction in hemolymph trehalose levels by enhancing trehalase activity (PMID: 25703302, 34564224). In *Spodoptera litura*, bovine insulin activated the PI3K/Akt/TOR signalling pathway and suppressed FoxO expression (PMID: 36005325), further supporting its role in modulating insect insulin signalling. Additionally, in *Tribolium castaneum* (Coleoptera), it was able to rescue phenotypes caused by ILP gene knockdown (PMID: 23754959), confirming its functional relevance.

Taken together, these findings support the use of bovine insulin as a practical and biologically meaningful tool to probe insulin-like pathways in insects.

2. The results section concerning the CvsNPF antibody doesn't quite make sense (lines 178-180). The experiment was set up as a test of whether CvsNPF is secreted into the host, and presumably interacts with host tissues. These data would be a major strength of the study, however, the only data presented are that teratocytes express the peptide.

Response: This is a good point. Combining the suggestion of the other reviewer, we use western blotting to verify the secretion of CvsNPF by teratocytes in parasitized host hemolymph using CvsNPF antibody. The result showed that CvsNPF antibody provoked immunological responses in the haemolymph of the parasitized host but not in that of the non-parasitized host (**Figure 4E**). The LC/MS analysis of the corresponding band detected a specific sequence of Cv-sNPF ([R].RNSLEGLRSSMSVDSPLYEHLIR.[K], positions in 48-70 of Cv-sNPF; PSM:1).

3. The molecular docking section (lines 286-308) should be more specific that the structures and interactions discussed are predictions rather than experimental data. In interpreted the manuscript as reporting experimentally derived structures of the NPF and NPFR proteins, with simulated docking. However, from the methods, I think that the structures are AlphaFold predictions instead.

Response: We apologize for any confusion caused. In the revised manuscript, we have clarified that the structures of sNPF and sNPFR were predicted using AlphaFold2 and that the docking results represent computational interaction models rather than experimentally determined structures. We hope this clarification addresses your concern.

4. I have some concerns about the immunofluorescence staining intensity analysis. The methods don't mention any form of normalization/control for fluorescence background. This is standard for fluorescence intensity analysis (i.e. the corrected total cell fluorescence [CTCF] approach), unless of course the data collected were photon counts, or similar. The methods section should be updated to address this issue, and the data reanalyzed if necessary (which should be fairly straightforward

using typical approaches).

Response: Thank you for catching that. You're right, we failed to mention the background correction process in our original description. In the revised manuscript, we have reanalyzed the fluorescence data following the corrected total cell fluorescence (CTCF) principle by subtracting the mean background fluorescence from the mean fluorescence intensity of each region of interest. We have updated the Methods section to clarify this procedure and have accordingly modified the figure data (**Fig. 5D and 5F**).

I noticed several typos throughout the manuscript, which I'm sure will be caught in proofreading, so no real issue at this stage but please make sure to double check everything, in particular references to figure panels.

Response: We have endeavoured to conduct a thorough review of the manuscript and have corrected all typos we could identify.

Referee #3:

In Wang et al., the authors focus on the relationship between the metabolism of the host *Plutella xylostella* and the parasitoid wasp *Cotesia vestalis* and report that parasitoid wasps regulate host metabolism through manipulation of the neuropeptide sNPF. Although some regulation of host behavior by parasitoids has been known, there are few examples of parasitoids' metabolic control through hijacking neuropeptide signaling, which readers will find interesting. This paper's message is derived from various experiments, including gene knockdown, peptide injections, and molecular simulations. However, some links between host trehalose metabolism and parasite development are missing. Moreover, the effects of neuropeptides on the host also need to be scrutinized in terms of parasite development and methods. Therefore, I argue for additional experiments (Major concern) and revisions (Minor concern).

Major concern

λ Host sugar metabolism and parasitoid wasp growth

1. It is unclear how the lack of an increase in blood trehalose in the host *Plutella xylostella* would affect the metabolism of the parasitoid wasp *Cotesia vestalis*. Since the decrease in trehalose due to pxGP knockdown is slight (roughly 20 µg/ µL hemolymph, it is even higher than non-parasite), it is unlikely that this slight decrease would affect the development of parasitoid wasps. Rather, parasitism or knockdown of pxGP may cause changes in the amounts of G6P or other intermediate metabolites of the glycolytic system, metabolites of the TCA circuit, and amino acids, which may affect the development of parasitoid wasps. To examine that trehalose is, in fact, the molecule that supports parasitoid bee growth, knockdown of TPP and TPS should be performed to investigate the relationship between trehalose synthesis and parasitoid wasp growth.

Response: Thank you for your valuable comments and suggestions. No homology of TPP was identified in the genome of *Plutella xylostella*. Thus, we knocked down *TPS* in parasitized host larvae. It turned out that interfering with the expression of *TPS* did not affect the trehalose level, nor compromise the development and fitness of wasp larvae (**Fig. R3**).

Regarding the slight decrease in trehalose levels following pxGP knockdown, we propose that multiple tissues (such as hemocytes or midgut) and pathways (including HK-mediated trehalose

production) contribute to parasitism-induced hypertrehalosemia. This compensatory mechanism may partially restore the trehalose level when GP was knocked down. Though we cannot exclude the possibility that G6P and other intermediate metabolites may influence host other nutrients which may also affect the development of parasitoid wasps, the link between GP and host circulating trehalose level is quite strong in several ways: 1) GP expression was significantly increased since early 4th instar of parasitized host larvae (**Fig. 1D**) and the expression pattern is quite similar with the increased dynamics of trehalose level (**Fig. 1B**), while TPS was induced since middle 4th instar. 2) The knockdown of GP decreased host trehalose level and influenced the carbohydrate concentration in wasp larvae.

Figure R3. The effect of TPS knockdown on trehalose synthesis and parasitoid wasp growth. Two-tailed unpaired Student's *t*-test (** $p < 0.01$, not shown, $p > 0.05$).

2. Concerning Fig. 2F, we have found that fecundity is reduced in female parasitoid wasps, but is this restored by sugar feeding, like the reduced starvation tolerance seen in males? It is necessary to determine whether the reduction of glycogen after eclosion is directly related to the number of eggs laid.

Response: Adult parasitoid wasps (both mated and virgin) used for parasitization were fed a sugar diet for approximately two days prior to the experiment (added in the legend of Fig. 2F). As a synovigenic species, *Cotesia vestalis* benefits from sugar or nectar sources, which positively influence mature egg number, fecundity, and parasitism rates (PMID: 37774133, 31517086, 32296099). Consequently, it is challenging to evaluate the impact of glycogen reduction after eclosion under starvation conditions. To address this, we added two extra experimental groups: females emerging from dsGP-treated hosts (dsGP♀) mated with males from dsGFP-treated hosts (dsGFP♂), denoted as the dsGP♀ × dsGFP♂ group, and the reciprocal cross, dsGP♂ × dsGFP♀. The dsGP♀ & dsGFP♂ group exhibited fecundity levels comparable to those of the control group, while the dsGP♂ & dsGFP♀ group showed a slight (but statistically insignificant) decrease in fecundity (**Fig. R4**). Comparable results were observed in wasps emerging from dssNPFR-treated hosts (**Fig. R5H**). These results suggest that reduced glycogen levels in male wasps may contribute to decreased fecundity. Combining the effect of reduced glycogen level on the survival of male wasps, we hypothesize that the number of eggs laid is largely related to the reduction of glycogen in

male offspring. However, the detailed mechanism underlying this phenomenon requires further investigation.

Figure R4. Number of eggs laid per hour by each female offspring emerged from dsGP-treated host larvae. Data are represented as mean \pm SEM. One-way ANOVA and Tukey's multiple comparison tests (* $p < 0.05$, ** $p < 0.01$, not shown $p > 0.05$).

λ Validation of sNPF and sNPFR signaling

3. There is no evidence in the current manuscript that hijacking sNPF/sNPFR signaling aids parasitic wasp development. Therefore, the success of parasitoid wasp parasitism against sNPFR knockdown hosts should be examined, as shown in Fig. 2.

Response: Thank you for highlighting the need to link the sNPF/sNPFR signalling to parasitic wasp development. Following successful sNPFR knockdown in parasitized host larvae, we conducted additional experiments to validate this connection. The results showed a significant decrease in host trehalose level post-sNPFR knockdown (**Fig. R5B**). Despite the unchanged pupation rate and eclosion duration, other parameters—including the eclosion, the fecundity and longevity of male wasps fed with water—were comparable between wasps emerging from dsNPFR-treated hosts and those from dsGP-treated hosts (**Fig. R5C-5L**, results added in manuscript). These results suggest that the sNPF/sNPFR signalling facilitates parasitic wasp development by modulating host trehalose levels.

Figure R5. The effect of sNPF/sNPFR signalling aids parasitic wasp development. (A) RNAi efficiency of dsPxsNPFR in parasitized host larvae (n = 4). dsPxsGFP was used as the control. (B) Circulating trehalose concentration from *C. vestalis* parasitized host larvae treated with dsPxsGFP or dsPxsNPFR (n=7). (C) head capsule width of wasp larvae from dsPxsNPFR-treated host larvae (n=50). (D-E) pupation rate (D, n = 3-4) and eclosion rate (E, n = 3) of wasp offspring. dsPxsNPFR. The total number of wasps for each treatment in (D and E) was shown on each bar graph. (F-G) Eclosion time of *C. vestalis* males (F, n = 144 for blue and n = 81 for red) and females (G, n = 137 for blue, n = 66 for red). (H) Number of eggs laid per hour by each virgin (left) or mated (right) female offspring. Adult parasitoid wasps (both mated and virgin) used for parasitization were fed a sugar diet for approximately two days prior to the experiment. (I,J) Survival curves of *C. vestalis* males (n = 32 for grey, n = 31 for red) and females (n = 30 for grey, n = 31 for red) fed with water. (K,L) Survival curves of *C. vestalis* males (n = 35 for grey, n = 35 for red) and females (n = 31 for grey, n = 37 for red) fed with sugar diet. Data are represented as mean ± SEM. Two-tailed unpaired Student's *t*-test for (A-E). One-way ANOVA and Tukey's multiple comparison tests for (F-H), and Kaplan-Meier log-rank test for (I-L). *** p < 0.01, not shown p > 0.05.

4. Since no validation of the CvsNPF antibody has been done, more data is needed to support the evidence that CvsNPF is expressed in Teratocytes. The expression of sNPF in Teratocytes should be validated by qPCR or Western blot using Teratocytes. In addition, I could not find any data on sNPF expression in teratocytes from the manuscript cited in L171-175. If the expression of sNPF were detected in a previous study (Gao et al., 2016), it would be preferable to reanalyze and show the data of sNPF expression from the data of previous studies.

Response: PCR and Western blot were used to detect CvsNPF expression and secretion by teratocytes, respectively (Fig. S3E and Fig. 4E). Furthermore, the LC/MS analysis of the corresponding band in Western blotting detected a specific sequence of CvsNPF ([R].RNSLEGLRSSMSVDSPEYELMIR.[K], positions in 48-70; PSM:1). Therefore, these results demonstrated that teratocytes secrete CvsNPF into host hemolymph.

5. Regarding Fig. 5E, looking at the source data for the image, there appears to be an increase in what looks like round cells expressing sNPF. Are these cells tracheal cells, hemocytes, or enteroendocrine cells? These cells should be mentioned and discussed to better understand the phenomenon. Furthermore, since antibody staining sometimes detects many nonspecific signals,

validation that the cells stained with antibodies are sNPF should be performed by the knockdown of sNPF.

6. The image in Fig. S6B suggests that sNPF acts autocrine on sNPFR in the trachea. Is hexokinase expressed in the intestinal trachea? It is worth discussing whether the trachea regulates whole-gut metabolism or an alternative possibility. As for the sNPFR antibodies, since antibodies against GPCRs tend to be challenging to produce, it would be desirable to validate whether the signals seen in Fig. S6B are due to sNPFR using Western blotting or sNPFR knockdown.

Response: As indicated in the text, the cells expressed sNPF in Fig. 5E are tracheal cells.

Hexokinase expressed in the intestinal trachea significantly increased after parasitization, confirming that sNPF acts autocrinally on sNPFR in the trachea (Fig. R6A). We further discuss the expression of sNPF and the potential role of tracheal sNPF/sNPFR signalling in regulating whole-gut metabolism (**Lines 377-381**).

The specificity of sNPF and sNPFR antibodies (generated against recombinant proteins) was verified by Western blotting and LC/MS analysis. Interestingly, both the sNPF and the sNPFR antibodies detected a band around 60 kDa in the midgut (4M) of host larvae (**Fig. R6B**).

Correspondingly, both Px-sNPF (Coverage 34%; PSMs: 4) and PxsNPFR ([Coverage 8%; PSMs:1) were identified in this bond (**Table R1**).

Figure R6. (A) Expression of *Hexokinase* (HK) is expressed in the intestinal trachea after parasitization. (B) Western blotting analysis of host midgut against anti-sNPF (left, 1:500) and anti-sNPFR (right, 1:500) antibodies. The HRP Goat Anti-Rabbit IgG (ABclonal, 1: 2500 dilution) was used as the secondary antibody.

Table R1. LC-MS result of sNPF and the sNPFR antibodies detected bond

Description	Coverage [%]	# Unique Peptides	MW [kDa]
Px-sNPF	34	4	19.6
Annotated Sequence	Modifications	# PSMs	Positions
[R].RSDPDMPPQAPLDEMEELLSLR.[D]		1	Px-sNPF [85-106]
[R].SDPDMPPQAPLDEMEELLSLR.[D]	2xOxidation [M5; M14]	1	Px-sNPF [86-106]
[R].SAETAVPHVFPQEEQDRSVR.[A]		1	Px-sNPF [120-139]
[R].SDNNMFLLPYESALPKDVK.[A]	1xOxidation [M5]	1	Px-sNPF [151-169]
PxsNPFR	8	1	50.2
Annotated Sequence	Modifications	# PSMs	Positions
[R].MLIAMVAIFGLSWLPLNINISSDFYSFAEDWR.[Y]	1xOxidation [M5]	1	PxsNPFR [253-285]

7. Since Px-sNPFs mature from a common prepro-sNPF via processing, the host cells should produce not only sNPF-1 but also sNPF-2 and 3 together. However, sNPF-2, 3 did not cause an increase in trehalose like sNPF-1 as in Fig.5. Can host sNPFs distinguish between sNPF-1, 2, and 3? Discussing the differences in the actions of sNPF-2, 3 as well as CvsNPF and sNPF-1 would enhance readers' understanding of neuropeptide actions in the future.

Response: We acknowledge that our original manuscript did not adequately address the functional differences among Px-sNPF-1, -2, and -3.

Two reasons may explain the differences in individual sNPF peptides' actions: (1) The variation may be attributed to differences in the binding affinities between the ligands and their respective receptors. (2) These differences are likely due to the differential interaction of the individual peptides with the host sNPFs, which activates distinct G protein signalling pathways. Furthermore, we employed structural prediction and receptor docking analysis to compare CvsNPF and PxsNPF-1. Although both peptides contain the conserved C-terminal RFamide motif, their overall 3D conformations and binding orientations within the receptor differ significantly. Related discussion has been updated in the revised manuscript (**Lines 419-436**).

Minor concern

8. Regarding the heatmap of RNA-seq in Fig. 3B, showing the variation of gene expression of NP and P between different tissues highlights the differences between tissues rather than between NP and P. The heatmap should be separated for each tissue to show the variation in gene expression between NP and P.

Response: Thank you for your valuable suggestion, which has helped improve the clarity of our data presentation. We have revised **Fig. 3B** to generate separate heat maps for each tissue. These updates allow for a clearer comparison of gene expression differences between the nonparasitized hosts (NP) and parasitized hosts (P) groups in each specific tissue context. This adjustment improves the interpretability of the data and more directly achieves the core objective of identifying differences between NP and P.

9. The authors examine the expression of InR in Fig.3D, and it appears to be upregulated in P compared to NP. This is not seen in the RNA-seq heatmap in Fig. 3B. Elevated InR expression is a typical phenotype of insulin resistance (Pasco and Leopold, 2012, PMID: 22567167), suggesting that hyperglycemia is causing insulin resistance in host larvae. Indeed, bovine insulin fails to cause a decrease in trehalose at P (Fig. 3F), which is also a typical phenotype of insulin resistance. Since insulin resistance further exacerbates hyperglycemia, insulin resistance may be involved in the host hyperglycemic state found by the authors, and the current claim of insulin signaling-independent needs to be changed.

Response: We sincerely appreciate the reviewers' valuable suggestions, which have helped improve the accuracy and depth of our discussion. After reanalyzing the heatmap data of RNA-seq, the upregulation of InR expression in the parasitized group was clearly demonstrated better (**Fig. 3B**), consistent with our qPCR results (**Fig. 3D**).

Though "insulin resistance" was mentioned in the Discussion, we failed to link it to parasitism-

induced hypertrehalosemia. Despite the presence of a typical insulin resistance phenotype, the unaltered GP expression in parasitized hosts treated with bovine insulin and AKH indicates that other signalling pathways are involved in its regulation (Fig. 3E and 3I). Furthermore, our data showed that the activation of the sNPF-sNPFR pathway by injection of sPxNPF-1 and sCvsNPF is sufficient to elevate trehalose levels in unparasitized larvae (Fig. 4F and 5B) and sCvsNPF injection elevated GP expression (Fig. 6C), while knockdown of sNPF or sNPFR decreased trehalose levels in parasitized larvae (Fig. 5I and Fig. R5B). Therefore, we have revised the manuscript text to clarify that the observed hypertrehalosemia is the result of the combined effects of insulin resistance and enhanced sNPF signalling.

10. Regarding Fig. 3C, are all bands of pAkt and AKT detecting PxAKT? AKT is around 60 kDa in many organisms, and whether the band in Fig. 3C, which is 70 kDa, detects AKT is questionable. The predicted Mw of a protein (XP_048483457.1) that appears to be an AKT homolog of *Plutella xylostella* is around 60 kDa. In some cases, such as in *Drosophila*, there is a Large isoform, and two bands are detected (see PMID: 20333234, Fig. 6), but it is necessary to check the blotting bands against the actual size of the protein.

Response: We appreciate your highlighting our oversight regarding the specificity of the AKT antibody. The previously used antibody (Phospho-Drosophila Akt (Ser505) Antibody #4054) was described to detect a band at approximately 65 kDa (<https://www.cellsignal.cn/products/primary-antibodies/phospho-drosophila-akt-ser505-antibody/4054>).

However, as this antibody is now discontinued, we devoted considerable effort to identifying a suitable replacement. The new AKT and pAKT antibodies we selected detect a band at approximately 60 kDa in both the fat body and midgut of *P. xylostella* larvae. This molecular weight corresponds to the expected size of PxAKT, which helps confirm the specificity of the new antibodies.

Figure R7. WB of Phospho-Drosophila Akt (Ser505) Antibody #4054.

11. Which tissue is used for qPCR data, such as Fig. 5A? If it is a whole body, it should be stated in the legend or the text. Also, it is better to describe the tissues used for the data and the instar of the larvae in the legend throughout the paper to help readers better understand the data.

Response: Thank you for your helpful suggestion. We have revised the figure legends to clearly indicate the tissue sources used for all qPCR data, including Fig. 5A, as well as the larval instar stage. These details have now been added consistently throughout the manuscript to improve

clarity for readers.

12. The method does not describe the solvent and volume of injection (nL~ μ L) for microinjection of sNPF. This information should be included for researchers who will conduct similar experiments.

Response: Thank you for pointing out this omission. We have updated the Methods section to include the injection details, specifying that sNPF was administered in a volume of 0.1 μ l per larva.

13. Legend: Fig. 3F and 3E are reversed.

Response: Corrected.

14. L574-576: Immuno blot data for CvsNPF does not appear to be included in the manuscript.

Response: Result of Immuno blot data for CvsNPF has been included in the revised Fig. 4E.

15. L710: The sequence used to create the sNPFR antibody should be described

Response: sNPFR antibody was generated using recombinant protein sNPFR (amino acids 1-42, **Line 750**).

16. Source data: Fig.3D-3F is missing

Response: Source data of Fig.3D-3F was added.

Dear Dr. Chen,

Thank you for submitting a revised version of your manuscript. It has now been seen by two of the original reviewers, and I have copied their comments below. As you can see, they are generally satisfied with the revisions and now recommend acceptance of the manuscript after incorporation of minor revisions.

Additionally, there are a few editorial points that need addressing before I can extend official acceptance of the manuscript:

1. Please remove figures from the manuscript text file.
2. Please correct the order and the headings of the manuscript sections to: Abstract / Keywords / Introduction / Results / Discussion / Methods / Data Availability / Acknowledgements / Disclosure and Competing Interests Statement / References / Figure Legends.
3. Please check if the correct email address has been provided for the author Min Shi, as our messages were returned.
4. In the Author Checklist file, please fill in the rows 99-101 and 108.
5. CRedit has replaced the traditional author contributions section because it offers a systematic, machine-readable author contributions format that allows for more effective research assessment. Please remove the Authors Contributions from the manuscript and use the free text boxes beneath each contributing author's name in our online submission system to add specific details on the author's contribution. More information is available in our guide to authors.
6. Please rename "Competing interests" section into "Disclosure and competing interests statement" (further info: <https://www.embopress.org/page/journal/14602075/authorguide#conflictsofinterest>)
7. Please rename datasets into Dataset EV1-2 throughout the manuscript. Please remove Datasets from the Appendix table of contents and add their legends directly to dataset files in a separate tab/worksheet.
8. Please enhance image resolution in the Appendix.
9. In the Appendix, please add page numbers to the table of contents. Please update the nomenclature to Appendix Figure S1-S9 and Appendix Table S1-S3 throughout, including the reference in the Reagents and Tools table.
10. In the Data Availability section, please add a resolvable link for PRJNA1073706 dataset. More information about the format of this section can be found here: <https://www.embopress.org/page/journal/14602075/authorguide#dataavailability>.
11. Please submit unmodified original image data as source data for figure panels 1F and 4H.
12. In our standard text plagiarism check, we noted that a sentence in the introduction appears identical to a previously published article (lines 31-32, please see in the attachment). Please consider rephrasing, if possible.
13. Our data editors have flagged the following issues in figure legends that need correcting:
 - Please provide the exact p values in the legends of figures 1B, D, E, G, H, I; 2B, D, E, F, G; 3D, E, F, G; 4A, B, C, E, G, I; 5A, B, D, F, G, H, I, J K, L; 6A, B, C, G, H.
14. Papers published in The EMBO Journal are accompanied online by a 'Synopsis' to enhance discoverability of the manuscript. It consists of A) a short (1-2 sentences) summary of the findings and their significance, B) 3-4 bullet points highlighting key results and C) a synopsis image that is 550x300-600 pixels large (width x height, jpeg or png format). You can either show a model or key data in the synopsis image. Please note that the image size is rather small and that text needs to be readable at the final size.

With best wishes,

Ieva

We realize that it is difficult to revise to a specific deadline. In the interest of protecting the conceptual advance provided by the

work, we recommend a revision within 3 months (25th Dec 2025). Please discuss the revision progress ahead of this time with the editor if you require more time to complete the revisions.

Referee #1:

The authors answered most of my major and minor concerns.

However, the western blots in the revised manuscript still need quantification in Figures 3 and 4.

Referee #3:

The authors have sincerely addressed each of my revision requests, conducted numerous additional experiments, and provided convincing data and explanations to support their claims. I am genuinely grateful for their thorough and thoughtful responses.

However, there remains one final point that the authors should clarify regarding the Western blot of sNPF. Although LC/MS confirmed that the bands indeed contained the target sequences of CvsNPF and PxsNPF, the positions of the sNPF bands in both Fig. 4E and Fig. R6B appear higher than expected. If CvsNPF is secreted as the full-length protein without the signal peptide, this might not be an issue. However, the detection of PxsNPF at approximately 60 kDa is surprising. Since these data are critical for demonstrating antibody specificity, a brief explanation is necessary to clarify why the bands appear at higher positions.

Furthermore, when LC/MS was used to analyze the bands, to what extent was sNPF detected? The contents of Table R1 are only mentioned in the Point-by-Point Response file. I believe that including data or methodology in the main manuscript explaining how sNPF was detected by LC/MS and to what degree it was detected would help readers better understand and appreciate the findings.

Referee #1:

The authors answered most of my major and minor concerns.

However, the western blots in the revised manuscript still need quantification in Figures 3 and 4.

Response: We appreciate the reviewer's positive feedback for the revision. In the previous revision, the western blot in Fig. 3 was quantified and related results were present in Fig. S3A. Regarding that in Fig. 4E, the purpose of the Western blot was to confirm the presence of CvsNPF in host hemolymph qualitatively rather than quantitatively. As shown in the results, nonparasitized hosts did not exhibit a specific band, making quantification and comparative analysis unfeasible.

Referee #3:

The authors have sincerely addressed each of my revision requests, conducted numerous additional experiments, and provided convincing data and explanations to support their claims. I am genuinely grateful for their thorough and thoughtful responses.

However, there remains one final point that the authors should clarify regarding the Western blot of sNPF. Although LC/MS confirmed that the bands indeed contained the target sequences of CvsNPF and PxsNPF, the positions of the sNPF bands in both Fig. 4E and Fig. R6B appear higher than expected. If CvsNPF is secreted as the full-length protein without the signal peptide, this might not be an issue. However, the detection of PxsNPF at approximately 60 kDa is surprising. Since these data are critical for

demonstrating antibody specificity, a brief explanation is necessary to clarify why the bands appear at higher positions.

Furthermore, when LC/MS was used to analyze the bands, to what extent was sNPF detected? The contents of Table R1 are only mentioned in the Point-by-Point Response file. I believe that including data or methodology in the main manuscript-explaining how sNPF was detected by LC/MS and to what degree it was detected-would help readers better understand and appreciate the findings.

Response: Thank for your positive feedback and valuable comments and suggestion. Actually, many protein bands are shifted in size compared to the predicted molecular weight, e.g. due to proteolytic processing and various post-translational modifications, including glycosylation (PMID: 30297845). Regarding CvsNPF, we added an explanation for the detected larger size band, possible due to post-translational modifications, such as glycosylation. And the related data was included (Lines 191-194).

The LC/MS analysis revealed that the PxsNPF fragments are peptides originating from regions downstream of cleavage sites (Figure R). As neuropeptides are usually proteolytically cleaved during secretion, we hypothesize that the detected band is a partially processed propeptide prior to cleavage. Similarly, the band detected by anti-PxsNPF is larger than expected size, may also represent an immature form of PxsNPF or a glycosylated form. Unfortunately, the two bands were found at very similar molecular weights, making them difficult to separate by SDS-PAGE. Related contents have been added into the Appendix Fig. S10 and Appendix Table S3. The details of the LC/MS analysis are described in the Methods section under "Western blotting and LC-MS/MS Analysis."

```

1      ATGGCGCGCAGCATGCTCGGCGTGGCGGGCGGGCGGGCGGGCGTGCGCCCTGGCCGTGCTGTGCGCGCTGCCACC
1      M A R S M L G V A A A A A A C A L A V L C A L P T
76     GGAAACGCGCAGGCGCTCTCACAGTATGACTCAGTGGCGCAGTCAGCGCAAGAGGCGGGCGGGCTGGGACGCACTC
26     G N A Q A L S Q Y D S V A Q S A Q E A A G W D A L
151    GGCGGGCTGTACGCTCTGCTGGCCCAACACGACGCGCTCGGAGGTCACGCGCTCGCACGCAAGTCCGCTCGCTCT
51     G G L Y A L L A Q H D A L G G H A L A R K S V R S
226    CCGTCGAGGCGGCTGAGATTTGGACGCGCTCCGACCCTGATATGCCGCGCAGGCTCCGCTGGACGAGATGGAG
76     P S R R L R F G R R S D P D M P P Q A P L D E M E
301    GAGCTGCTGCTCGCTGCGGGACACGCGCCAGCCCGTGGCGCTGCGCTTCGGCCGCGTTCGGCCGAGACCGCGTG
101    E L L S L R D T R Q P V R L R F G R R S A E T A V
376    CCGCACGCTTCCCGCAGGAGGAGCAAGACCCTCGGTGCGCGCGCCCTCCATGCGCCTGCGCTTCGGCCGCGC
126    P H V F P O E E O D R S V R A P S M R L R F G R R
451    TCCGACAACAACATGTTCTACTGCCCTACGAGTCCGGCTCCTCCCAAGGACGTGAAAGCTAGTGGAGGCGAAGAC
151    S D N N M F L L P Y E S A L P K D V K A S G G E D
526    GACCGGACACAGGACTAA
176    D R T Q D *

```

Fig. R. PxsNPF prepropeptide sequence which is flanked by monobasic or dibasic cleavage sites and followed by 3 amidation signals. Underlined: signal peptide; bold: predicted mature peptide sequence; light grey: cleavage sites and dark grey: amidation; red underlined: detected PSMs.

Dear Dr. Chen,

Thank you for addressing the final editorial requests in the revised manuscript. I am now pleased to inform you that your manuscript has been accepted for publication in the EMBO Journal. Congratulations on a nice study!

Before we forward your manuscript to our publishers, we would like to propose some edits in the manuscript title, abstract and synopsis (please see below and in the attached file). I have also written a short blurb that will accompany the title of your manuscript in our online system. Please take a look and let me know if any corrections are needed.

New title:

Dual interference with host neuropeptide signaling allows parasitoid wasp to hijack host sugar metabolism

Blurb:

sNPF derived from both the host and the parasite induce trehalose accumulation by targeting distinct host tissues and signalling pathways.

Synopsis:

Carbohydrate metabolism influences the outcome of host-parasite relationships. This study shows that insect short neuropeptide F (sNPF) induces trehalose accumulation during the interaction between the parasitoid wasp *Cotesia vestalis* and its host, the moth *Plutella xylostella*.

- sNPFs produced by both the host and the parasitoid act as trehalose-increasing hormones.
- Parasitoid-derived sNPF promotes glycogenolysis in the host fat body via the host sNPF receptor.
- Parasitoid-derived virus activates host sNPF expression, stimulating glycolysis in the midgut.
- Host and parasite sNPFs exhibit differing receptor affinities and stimulate distinct downstream signalling pathways.

Please note that it is The EMBO Journal policy for the transcript of the editorial process (containing referee reports and your response letters) to be published as an online supplement to each paper. If you prefer removal of any referee-only figures included in the point-by-point response(s), e.g. because they may still be used for future publication or because they have been reproduced from published work by others, please do let us know immediately via response email.

More information is available here: https://www.embopress.org/transparent-process#Review_Process

If you have any questions, please do not hesitate to contact the Editorial Office or me directly. Thank you for your contribution to The EMBO Journal!

With best wishes,

Ieva
